# Efficient Generative Modeling beyond Memoryless Diffusion via Adjoint Schrödinger Bridge Matching

Jeongwoo Shin [1]   Jinhwan Sul [2]   Joonseok Lee [†1]   Jaewoong Choi [†3]   Jaemoo Choi [†1 2]

## Abstract

Diffusion models often yield highly curved trajectories and noisy score targets due to an uninformative, memoryless forward process that induces independent data-noise coupling. We propose **Adjoint Schrödinger Bridge Matching (ASBM)**, a generative modeling framework that recovers optimal trajectories in high dimensions via two stages. First, we view the Schrödinger Bridge (SB) forward dynamic as a coupling construction problem and learn it through a data-to-energy sampling perspective that transports data to an energy-defined prior. Then, we learn the backward generative dynamic with a simple matching loss supervised by the induced optimal coupling. By operating in a non-memoryless regime, ASBM produces significantly straighter and more efficient sampling paths. Compared to prior works, ASBM scales to high-dimensional data with notably improved stability and efficiency. Extensive experiments on image generation show that ASBM improves fidelity with fewer sampling steps. We further showcase the effectiveness of our optimal trajectory via distillation to a one-step generator.

## 1. Introduction

Generative modeling aims to sample from a data distribution $p_{\text{data}}$ by transforming a simple prior distribution $p_{\text{prior}}$ (*e.g.*, Gaussian). Diffusion models achieve this by learning continuous dynamic based on Stochastic Differential Equation (SDE) that connect $p_{\text{prior}}$ and $p_{\text{data}}$ (Song et al., 2021b; Ho et al., 2020; Song et al., 2021a). While highly successful, these methods face two significant limitations (Song et al.,

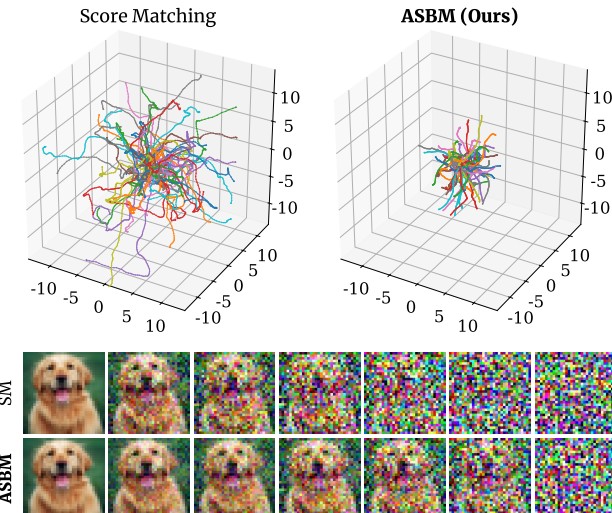

Figure 1. **Generation trajectory of score matching and ASBM.** *Top*: Backward drift accumulated over time in pixel level. *Bottom*: Denoising path in image level. ASBM shows significantly smaller transport cost with straighter path, leading to efficient generation.

2021b; Karras et al., 2022). First, the learned trajectories are often highly curved, requiring a large Number of Function Evaluations (NFEs) at sample generation. Second, their training objectives typically rely on independent endpoint pairing $(X_0, X_1) \sim p_{\text{data}} \times p_{\text{prior}}$, which yields noisy training targets and slow convergence. This independent coupling is induced by the *memoryless forward process* (Domingo-Enrich et al., 2025) (cf. Eq. (8)).

Optimal Transport (OT) (Villani et al., 2008; Peyré et al., 2019) provides a powerful alternative by seeking an *optimal coupling* between $X_0 \sim p_{\text{data}}$ and $X_1 \sim p_{\text{prior}}$ that minimizes a transport cost. Among various OT formulations, the Schrödinger Bridge (SB) problem (Schrödinger, 1931; Léonard, 2014; Chen et al., 2016; 2021; De Bortoli et al., 2021) is particularly relevant to diffusion models, because the optimal coupling is represented by a pair of consistent forward-backward SDE dynamics. Unlike the trivial independent coupling in standard diffusion, the cost-minimizing principle of SB induces the shortest path between $p_{\text{data}}$ and $p_{\text{prior}}$. By generalizing the memoryless forward process into the *non-memoryless* regime, SB can achieve optimal trajec-

[1]Seoul National University, Seoul, Korea [2]Geogia Institute of Technology, Atlanta, GA, USA [3]Sungkyunkwan University, Seoul, Korea. Correspondence to: Joonseok Lee <joonseok@snu.ac.kr>, Jaewoong Choi <jaewoongchoi@skku.edu>, Jaemoo Choi <jaemoo.choi@gatech.edu>.

*Proceedings of the 43rd International Conference on Machine Learning*, Seoul, South Korea. PMLR 306, 2026. Copyright 2026 by the author(s).

tories that are straighter than those obtained from independent endpoint pairing, leading to less NFEs for generation.

However, realizing the non-memoryless SB in high-dimensional settings, such as images, remains challenging. Existing SB-inspired generative methods (Chen et al., 2022; Deng et al., 2024) often resort to independent pairing $(X_0, X_1) \sim p_{\text{data}} \times p_{\text{prior}}$, or perform auxiliary pretraining with empirical bridge matching (Shi et al., 2023), limiting primary benefits of the optimal trajectories. Furthermore, they typically use forward–backward alternating training, requiring bidirectional trajectory rollouts to supervise each other. In practice, this bidirectional supervision is noisy and often leads to inconsistent dynamics that do not correspond to a single optimal path measure, which weakens the intended OT property and reduces sampling efficiency.

In this paper, we introduce *Adjoint Schrödinger Bridge Matching (ASBM)*, an SB-based generative modeling framework that efficiently learns organized trajectories under the informative endpoint couplings with highly stable convergence. We decompose generative modeling via Schrödinger Bridges into two simple subproblems: (i) constructing the endpoint optimal coupling using data-to-prior forward dynamic, and (ii) optimizing the backward dynamic with a simple matching loss supervised by the resulting optimal coupling. At stage (i), we view the SB problem as a Stochastic Optimal Control (SOC)-based sampling problem (Zhang & Chen, 2022; Vargas et al., 2023; Havens et al., 2025; Liu et al., 2025), which learns the optimal control that transports initial data $p_{\text{data}}$ to desired Boltzmann distribution, *i.e.*, unnormalized density with a known energy function. Since forward dynamic in SB is the transport from samplable distribution, *i.e.*, $p_{\text{data}}$, to energy-known $p_{\text{prior}}$, *e.g.*, Gaussian, the problem can be reformulated as a data-to-energy sampling problem and efficiently solved. At stage (ii), we use the learned forward process to obtain the optimal SB coupling $(X_0, X_1)$ and train the backward dynamic with a simple matching loss. Interestingly, our approach also recovers the standard diffusion model as a special case with a specific forward dynamic which yields independent endpoint coupling without training (See Sec. 3.1).

Our design brings three key advantages. First, ASBM requires only the forward simulation at training. Since it transports from data to a simple prior, it yields stable and fast optimization, requiring dramatically fewer NFEs (*e.g.*, 20 vs. 100–200 in prior work) to construct endpoint couplings, sufficiently with a lighter network. This stable forward training produces higher-quality optimal couplings, enabling more informative supervision for backward dynamic. Second, since optimal coupling is induced by learned forward dynamic, we can optimize backward dynamic via simple matching loss which converges fast with high stability. Third, ASBM's straighter trajectory results in improved

generative performance with lower NFE compared to prior SB-based generative models and diffusion model. We further showcase the effectiveness of our efficient trajectory via distillation to one-step generator, with better performance and mode coverage.

Our contributions are three-fold:

- We propose *Adjoint Schrödinger Bridge Matching (ASBM)*, which learns *optimal trajectory* in significantly *efficient and stable* manner through our novel perspective on Schrödinger Bridge optimization.
- ASBM achieves superior performance over diffusion model and prior SB methods on image generation with faster sampling (low NFE) and better fidelity.
- Leveraging the efficient trajectory of ASBM, we improve the sample quality and *mode coverage* in distillation task, compared to score-based distillation.

## 2. Background

**Diffusion Models (DMs)** (Song et al., 2021b; Ho et al., 2020) learn to sample from a target data distribution $p_{\text{data}}$ by reversing a fixed forward noising process. This forward process is designed to describe the stochastic dynamic from the data distribution $p_{\text{data}}$ to the prior distribution $p_{\text{prior}}$, typically a Gaussian. Specifically, consider a forward Stochastic Differential Equation (SDE) for $X_t \in \mathbb{R}^d$:

$$\mathrm{d}X_t = f_t^{\text{DM}}(X_t)\,\mathrm{d}t + \sigma_t^{\text{DM}}\mathrm{d}W_t, \qquad X_0 \sim p_{\text{data}}, \quad (1)$$

where $f^{\text{DM}} : [0, 1] \times \mathbb{R}^d \to \mathbb{R}^d$ is the base drift and $\sigma^{\text{DM}} : [0, 1] \to \mathbb{R}_{>0}$ is the noise schedule. This forward SDE has a corresponding reverse-time SDE (Song et al., 2021b) that follows the same stochastic dynamic backward in time:

$$\mathrm{d}X_t = \left[ f_t^{\text{DM}}(X_t) - \sigma_t^{\text{DM}^2} \nabla \log p_t(X_t) \right] \mathrm{d}t + \sigma_t^{\text{DM}} \mathrm{d}W_t,$$

for $X_1 \sim p_1$, where $\nabla_x \log p_t(x)$ is a marginal score and $p_1 \approx p_{\text{prior}}$ under a proper noise schedule in Eq. (1). DMs estimate this marginal score via conditional score matching (Song et al., 2021b; Ho et al., 2020) and generate samples by simulating the reverse SDE with an approximate score $s_t$ starting from $p_{\text{prior}}$:

$$\min_s \mathbb{E}_{p_{t|0}, X_0 \sim p_{\text{data}}} \left[ \left\| s_t(X_t) - \nabla_{x_t} \log p(X_t \mid X_0) \right\|^2 \right], \quad (2)$$

where $s : [0, 1] \times \mathbb{R}^d \to \mathbb{R}^d$ is a score function to approximate the marginal score.

**Schrödinger Bridge.** Consider a controlled SDE as

$$\mathrm{d}X_t = \left[ f_t(X_t) + \sigma_t u_t^\theta(X_t) \right] \mathrm{d}t + \sigma_t \,\mathrm{d}W_t, \; X_0 \sim p_{\text{data}}, \quad (3)$$

where $u^\theta : [0, 1] \times \mathbb{R}^d \to \mathbb{R}^d$ is a parameterized forward control. $p^u$ is the path measure induced by controlled SDE

in Eq. (3), and the base path measure $p^{\text{base}}$ is induced by the uncontrolled base SDE, *i.e.*, by setting $u \equiv 0$ in Eq. (3).

Schrödinger Bridge (SB) problem finds a path measure $p^u$ that matches both endpoint marginals, $p_{\text{data}}$ and $p_{\text{prior}}$, while minimizing the KL divergence relative to a reference base process $p^{\text{base}}$ (Schrödinger, 1931; Léonard, 2014):

$$\min_u D_{\text{KL}}\big(p^u \,\|\, p^{\text{base}}\big) = \mathbb{E}_{p^u}\left[\int_0^1 \frac{1}{2}\,\|u_t^\theta(X_t)\|^2\,dt\right], \quad (4)$$

subject to Eq. (3) and $X_1 \sim p_{\text{prior}}$. The optimal bridge admits a reverse-time SDE representation with the same boundary constraints:

$$\mathrm{d}X_t = \big[f_t(X_t) - \sigma_t v_t^\phi(X_t)\big]\,\mathrm{d}t + \sigma_t\,\mathrm{d}W_t,\ \ X_1 \sim p_{\text{prior}}, \quad (5)$$

where $v_t^\phi(x) : [0,1] \times \mathbb{R}^d \to \mathbb{R}^d$ is a parameterized backward control. Optimal path measure $p^\star$ can be induced by both direction with optimal controls $u_t^\star$ or $v_t^\star$.

**Reciprocal Property of SB.** Optimal path measure $p^\star$ follows a *reciprocal process* (Léonard et al., 2014):

$$p^\star(X_t) = p^{\text{base}}(X_t|X_0, X_1)\,p^\star(X_0, X_1). \quad (6)$$

This representation indicates that the optimal path measure is characterized by the optimal coupling $p^\star(X_0, X_1)$. Given the coupling of two terminal distributions, the intermediate distribution can be constructed from the base process.

## 3. Method

We introduce a generative model that generalizes the standard **Diffusion Models (DMs)** into a **non-memoryless regime** by leveraging the Schrödinger Bridge (SB) problem. Our approach addresses the fundamental inefficiencies of diffusion models, *i.e.*, curved trajectories and noisy training targets, by explicitly recovering optimal transport couplings.

### 3.1. Diffusion Models as Memoryless SB

**Bridge Matching.** As shown in Shi et al. (2023), given the optimal coupling $(X_0, X_1) \sim p_{0,1}^\star$, we could obtain the optimal backward control $v_t^\star$ by solving

$$\min_\phi \mathbb{E}_{p_{t|0,1}^{\text{base}}\,p_{0,1}^\star}\left[\left\|v_t^\phi(X_t) - \sigma_t\nabla\log p_{t|0}^{\text{base}}(X_t \mid X_0)\right\|^2\right]. \quad (7)$$

This property implies that if the optimal joint distribution $p^\star(X_0, X_1)$ is accessible, we can optimize the backward control $v_t^\phi$ via bridge matching (Liu et al., 2023a).

**Connection between DMs and SB.** In DMs, we set the base drift and diffusion term $(f^{\text{DM}}, \sigma^{\text{DM}})$ so that the $X_1$ sampled from $p_{1|0}^{\text{base}}(\cdot|X_0)$ is almost independent to $X_0$, *i.e.*,

$$p_{0,1}^{\text{base}}(X_0, X_1) \overset{\text{memoryless}}{:=} p_0^{\text{base}}(X_0)\,p_1^{\text{base}}(X_1). \quad (8)$$

We denote this condition as a *memoryless condition*, and underlying dynamics as a *memoryless dynamics*. Then, the following theorem holds:

**Proposition 3.1.** *If the base path measure $p^{base}$ is memoryless, then the optimal path measure $p^\star$ of the SB problem is also memoryless, i.e.,*

$$p^\star(X_0, X_1) \overset{\text{memoryless}}{=} p_{\text{data}}(X_0)\,p_{\text{prior}}(X_1). \quad (9)$$

Consequently, under the memoryless condition, the expectation in Eq. (7) reduces to

$$p_{t|0,1}^{\text{base}} \cdot p_{0,1}^\star \ \ \Leftrightarrow\ \ p_{t|0,1}^{\text{base}} \cdot p_{\text{prior}} \cdot p_{\text{data}} \ \ \Leftrightarrow\ \ p_{t|0}^{\text{base}} \cdot p_{\text{data}}, \quad (10)$$

which is exactly the same form as the score matching objective in Eq. (2). This indicates that the diffusion model is a special case of SB, where the forward dynamic in Eq. (1) is a fixed memoryless base dynamic.

**Limitation of Memoryless Base SDE.** This interpretation shows the fundamental inefficiency of the score matching. In the memoryless regime, the injection of *massive* noise makes the matching target $\nabla_{x_t}\log p^{\text{base}}(X_t \mid X_0)$ highly stochastic. This leads to a slow convergence and highly curved backward path, leading to numerous function evaluations for generating high-quality samples (Lipman et al., 2023; Karras et al., 2022). In other words, as endpoint couplings are independent from each other, it is not informative to learn effective generation path. Therefore, we propose a method to obtain an informative optimal coupling $p^\star(X_0, X_1)$ to more efficiently supervise our backward bridge matching to learn straighter generation path.

### 3.2. Adjoint Schrödinger Bridge Matching

To break through the limitations of the memoryless dynamics, we adopt a *non-memoryless* base SDE to induce informative optimal couplings, and thereby learn efficient generation trajectory. Our core contribution is a *decoupled optimization of forward-backward dynamics* in a two-stage process: 1) **Optimal Coupling Construction** for the forward process by a novel interpretation of it as a *data-to-energy* sampling problem, and 2) **Backward Dynamic Optimization** via simple matching loss using *reciprocal process* under optimal coupling $p^\star(X_0, X_1)$.

**Optimal Coupling Construction.** Since non-memoryless base SDE no longer transports $X_0$ to $p_{\text{prior}}$ on its own, we need to optimize additional forward control $u_t^\theta$ in Eq. (3) to enforce the terminal marginal $p_{\text{prior}}$. Stochastic Optimal Control (SOC)-based sampling problem (Zhang & Chen, 2022; Vargas et al., 2023; Havens et al., 2025; Liu et al., 2025) finds the optimal control which tilts the base SDE to transport a samplable distribution to a target Boltzmann distribution, which is known up to its energy function.

Our **novel perspective** is that, within the generative modeling framework, the forward dynamic in the SB problem (3)

can be seen as a *data-to-energy* sampling problem. In this viewpoint, the forward process transports from an empirical distribution $p_{\text{data}}$ to an energy-known distribution $p_{\text{prior}}$, *e.g.*, Gaussian. This is highly beneficial because the energy gradient provides a dense, point-wise characterization of $p_{\text{prior}}$, whereas empirical supervision relies on finite samples and can only specify the target distribution through sparse Monte Carlo estimates. This reformulation allows us to isolate the coupling construction part from the unstable alternating optimization of forward-backward system.

Then, our first goal is finding the *optimal control* $u_t^{\star}$ of sampling problem. Under the SB optimality (Pavon & Wakolbinger, 1991; Chen et al., 2021; Caluya & Halder, 2021), the optimal controls can be characterized by

$$u_t^{\star}(x) = \sigma_t \nabla_x \log \varphi_t(x), \quad v_t^{\star}(x) = \sigma_t \nabla_x \log \hat{\varphi}_t(x), \quad (11)$$

where $\varphi_t, \hat{\varphi}_t \in C^{1,2}([0,1], \mathbb{R}^d)$ are SB potentials satisfying

$$\varphi_t(x) = \int p_{1|t}^{\text{base}}(y \mid x) \varphi_1(y) \, dy, \quad \varphi_0(x) \hat{\varphi}_0(x) = p_{\text{prior}}(x),$$

$$\hat{\varphi}_t(x) = \int p_{t|0}^{\text{base}}(x \mid y) \hat{\varphi}_0(y) \, dy, \quad \varphi_1(x) \hat{\varphi}_1(x) = p_{\text{data}}(x).$$

Optimal controls in Eq. (11) are typically obtained by alternating optimization of $\varphi_t$ and $\hat{\varphi}_t$ (Fortet, 1940; Kullback, 1968; Cuturi, 2013; Shi et al., 2023). However, since we focus on *data-to-energy sampling* framework, we **only** need the forward optimal control $u_t^{\star}$, which can be learned by alternating Adjoint Matching (AM) (12) and Corrector Matching (CM) (13) (Liu et al., 2025):

$$\min_{\theta} \; \mathbb{E}_{p_{t|0,1}^{\text{base}}, p_{0,1}^{\bar{u}^{\theta}}} \left[ \left\| u_t^{\theta}(X_t) + \left( \sigma_t \nabla E + \bar{v}_1^{\phi} \right)(X_1) \right\|^2 \right], \quad (12)$$

$$\min_{\phi} \; \mathbb{E}_{p_{0,1}^{\bar{u}^{\theta}}} \left[ \left\| v_1^{\phi}(X_1) - \sigma_1 \nabla_{x_1} \log p^{\text{base}}(X_1 \mid X_0) \right\|^2 \right], \quad (13)$$

where $\bar{u} = \text{stopgrad}(u)$ and $\bar{v} = \text{stopgrad}(v)$, and we define the energy $E$ by $p_{\text{prior}} \propto \exp(-E(x))$. Note that CM is same as bridge matching only at $t = 1$. See Appendix B for more details.

By training forward dynamic via SOC framework without any reliance on backward dynamic, we can optimize the forward control to approximate the optimal coupling with **high stability**. This also allows us to simulate endpoint pairs via only the forward dynamic, which has various advantages. Since forward control learns dynamic from complicated data space to simple prior, it is much **easier to learn** compared to backward dynamic, leading to **lighter model capacity** (Appendix E) with **fast convergence** (Sec. 4.5). Most importantly, this high stability of our optimization scheme allows us to adopt **non-memoryless** base SDE which requires more complicated training compared to the memoryless one.

Upon convergence, our forward dynamic (3) approximates optimal couplings $p^{\star}(X_0, X_1)$. It is important to note that

---

**Algorithm 1** ASBM optimization with VP base SDE

**Require:** $X_0 \sim p_{\text{data}}$; $p_{\text{prior}}(x) \propto e^{-E(x)}$; forward control $u_t^{\theta}(\cdot)$, backward control $v_t^{\phi}(\cdot)$; steps $N_1, N_2$.
1: **Base SDE:** $f_t(x) := -\frac{1}{2}\beta_t x$, $\sigma_t := \sqrt{\beta_t}$, with $\beta_t := (1-t)\beta_{\max} + t\beta_{\min}$.
2: **Reciprocal sampler:** Given $\kappa_t := \exp\left(-\frac{1}{2}\int_t^1 \beta_\tau \, d\tau\right)$, $\bar{\kappa}_t := \exp\left(-\frac{1}{2}\int_0^t \beta_\tau \, d\tau\right)$, $X_t \sim p_{t|0,1}^{\text{base}}(\cdot \mid X_0, X_1) = \mathcal{N}(\mu_t, \Sigma_t I)$ where,
$$\mu_t = \frac{\bar{\kappa}_t(1 - \kappa_t^2)}{1 - \bar{\kappa}_1^2} X_0 + \frac{\kappa_t(1 - \bar{\kappa}_t^2)}{1 - \bar{\kappa}_1^2} X_1,$$
$$\Sigma_t = \frac{(1 - \kappa_t^2)(1 - \bar{\kappa}_t^2)}{1 - \bar{\kappa}_1^2}.$$
3: **Stage 1:** optimize $u_t^{\theta}$ via SOC.
4: **for** $i = 1$ to $N_1$ **do**
5:     Update $\theta$ via Eq. (12).
6:     Update $\phi$ at $t=1$ via Eq. (13).
7: **end for**
8: **Stage 2:** optimize $v_t^{\phi}$ via BM with reciprocal sampler, under $p_{0,1}^{\bar{u}^{\theta}}(X_0, X_1) \approx p_{0,1}^{\star}(X_0, X_1)$.
9: **for** $i = 1$ to $N_2$ **do**
10:     Update $\phi$ via Eq. (14).
11: **end for**

---

our $p^{\star}(X_0, X_1)$ is **optimal coupling**, since our framework minimizes the transport cost (4) while considering the non-trivial correlation between $X_0$ and $X_1$ via the non-memoryless condition. Intuitively, since non-memoryless base SDE injects smaller noise compared to the forward process in standard diffusion, *i.e.*, $\sigma_t \ll \sigma_t^{\text{DM}}$, transport cost minimization leads to significantly straighter trajectories, allowing considerably **low NFEs** (Sec. 4.4) for simulating endpoint pair. This property is fundamentally unattainable under the standard memoryless forward dynamic in diffusion models, relying on independent endpoint pairings.

**Backward Dynamic Optimization.** Given the optimal joint $p^{u^{\theta}}(X_0, X_1)$ induced by the optimized forward dynamic (3), we supervise our backward dynamic by bridge matching in Eq. (7) under these couplings:

$$\min_{\phi} \; \mathbb{E}_{p_{t|0,1}^{\text{base}}, p_{0,1}^{\bar{u}^{\theta}}} \left[ \left\| v_t^{\phi}(X_t) - \sigma_t \nabla_{x_t} \log p^{\text{base}}(X_t \mid X_0) \right\|^2 \right]. \quad (14)$$

With direct supervision under an optimal coupling, backward training converges much faster than memoryless diffusion baselines. Moreover, the two-stage design enables principled use of the **reciprocal process**, which is exact only when the optimal endpoint coupling $p^{\star}(X_0, X_1)$ is available. In contrast, prior methods typically lack in optimal coupling and thus rely on alternating forward-backward optimization with reciprocal process of imperfect endpoints. Our full algorithm is described in Algorithm 1.

**Advantages of ASBM.** Previous methods (De Bortoli et al., 2021; Chen et al., 2022; Shi et al., 2023; Liu et al., 2022;

2024a; Chen et al., 2023) address the SB problem by alternating the optimization of forward and backward dynamics, repeatedly generating trajectories using the current forward (resp. backward) model to provide supervision for updating the backward (resp. forward) dynamic. Despite extensive prior work, scaling such alternating schemes to high-dimensional settings remains challenging. These approaches implicitly assume that trajectories generated by one direction provide sufficiently informative supervision for optimizing the opposite direction. However, in generative modeling, learning the backward dynamic, from a simple prior to a complex data distribution, is particularly difficult. As a result, inaccurate trajectories learned at training can destabilize the optimization of the counterpart dynamic and lead to error accumulation, ultimately yielding mismatched forward and backward processes and non-optimal couplings (see Sec. 4.2). To mitigate this, prior methods either resort to memoryless base dynamics, which diminishes the benefits of SBs, or rely on pretraining stages using independent couplings, which do not fully align with the theoretical formulation and may lead to practical instability.

In summary, our two-stage optimization completely resolves these issues. By isolating the forward optimization as data-to-energy sampling problem, we obtain optimal coupling with (i) high stability, (ii) low NFEs, and (iii) light model capacity under (iv) non-memoryless condition. Our backward optimization also becomes considerably simpler and more stable through the bridge matching loss with exact reciprocal process under optimal coupling.

### 3.3. Distillation to One-Step Generator

To further verify the effectiveness of the optimal trajectory of our method, we introduce a *data-free* distillation method for one-step generator within the SB framework. This part demonstrates the inherent strengths of our approach: since the learned generative paths are significantly more organized than those in standard diffusion models, ASBM provides a more suitable and efficient foundation for distillation.

**Distillation in Control Space.** We aim to distill our learned backward control $v_t^\phi$ to a one-step generator $G^\psi : \mathbb{R}^d \to \mathbb{R}^d$. Upon successful training, we are given the learned backward controlled SDE (5), which approximately reaches the data marginal $p_{\text{data}}$ at $t = 0$. We denote the path measure induced from the backward dynamic as $p^\phi$. Let

$$p_{0,1}^\psi := \text{Law}\big((G^\psi(X_1), X_1)\big) \qquad (15)$$

denote the joint distribution induced by $X_1 \sim p_{\text{prior}}$ and $z \sim \mathcal{N}(0, I)$, where $G^\psi$ is a one-step generator mapping the prior sample $X_1$ to a data sample $X_0 = G^\psi(X_1)$.

We further extend the notation $p_{0,1}^\psi$ to a full path measure $p^\psi$ on $[0, 1] \times \mathbb{R}^d$, which represents an (implicit) stochastic process whose endpoint coupling is given by $p_{0,1}^\psi$. Our goal

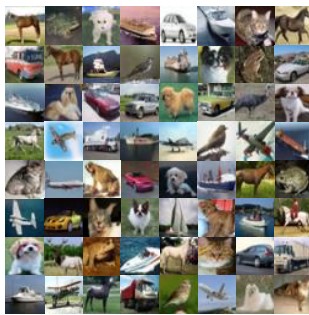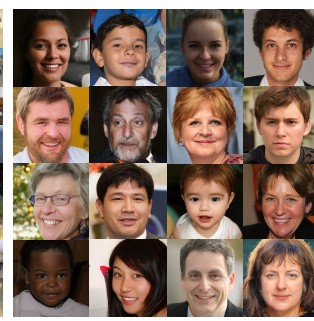

CIFAR-10 (32 × 32)  FFHQ (256 × 256)

*Figure 2.* Generated samples from ASBM on pixel space (CIFAR-10) and on latent space (FFHQ).

is to match this path measure with a target path measure $p^\phi$, by minimizing the path-space KL divergence $D_{\text{KL}}(p^\psi \| p^\phi)$. In particular, achieving $p^\psi \approx p^\phi$ ensures that the induced marginal satisfies $p_0^\psi = p_{\text{data}}$, *i.e.*, $G^\psi$ successfully transports the prior distribution to the data distribution.

Since the path measure $p^\psi$ induced by $G^\psi$ is not explicitly tractable, we introduce a control-based parametrization to approximate it. Specifically, we consider a controlled diffusion whose path measure $p^\xi$ is induced by

$$dX_t = \big[f_t(X_t) - \sigma_t v_t^\xi(X_t)\big] dt + \sigma_t dW_t, \quad X_1 \sim p_{\text{prior}}, \quad (16)$$

where $v^\xi : [0, 1] \times \mathbb{R}^d \to \mathbb{R}^d$ is a parameterized control. The role of $v^\xi$ is to parameterize $p^\xi$ that approximates the implicit path measure $p^\psi$. As a result, we need to alternately optimize $G^\psi$ and $v^\xi$ to reflect continuously changing $p_{0,1}^\psi$.

Given the current one-step generator $G^\psi$, we aim to learn control $v^\xi$, such that the corresponding path measure $p^\xi$ correctly estimates $p^\psi$. Given endpoint pairs $(X_0, X_1) \sim p_{0,1}^\psi$, we utilize bridge matching to update $v^\xi$:

$$\min_\xi \mathbb{E}_{p_{t|0,1}^{\text{base}}, p_{0,1}^\psi}\Big[\big\|v_t^\xi(X_t) - \sigma_t \nabla \log p_{t|0}^{\text{base}}(X_t \mid X_0)\big\|^2\Big]. \quad (17)$$

This objective encourages $v^\xi$ to approximate the score of the base bridge conditioned on $X_0$, thereby matching the induced path measure $p^\xi$ to $p^\psi$.

Finally, we update the generator $G^\psi$ by minimizing the discrepancy between the learned control $v^\xi$ and the target control $v^\phi$. Concretely, we solve

$$\min_\psi \ \mathbb{E}_{p_{t|0,1}^{\text{base}}, p_{0,1}^\psi}\Big[\big\|\bar{v}_t^\xi(X_t) - \bar{v}_t^\phi(X_t)\big\|^2\Big], \qquad (18)$$

which can be derived from a KL minimization that aligns the $p^\xi$ to $p^\psi$ by Girsanov's theorem (Särkkä & Solin, 2019) (See Appendix C for detailed derivation). With memoryless forward base process, our SB distillation recovers the score distillation (Poole et al., 2023), following Eq. (10).

| Method | OC | No PT | FB | FID ↓ |
|---|---|---|---|---|
| *Refinement method* | | | | |
| DOT (Tanaka, 2019) | ✓ | ✗ | – | 15.78 |
| DGFLOW (Ansari et al., 2021) | ✓ | ✗ | – | 9.63 |
| *Flow-based method* | | | | |
| FM (Lipman et al., 2023) | ✗ | ✓ | ✓ | 6.35 |
| Rectified Flow (Liu et al., 2023b) | ✓ | ✓ | ✗ | 6.01 |
| OT-CFM (Tong et al., 2024a) | ✓ | ✓ | ✗ | 4.15 |
| *Memoryless SDE* | | | | |
| Score SDE (Song et al., 2021b) | ✗ | ✓ | ✓ | 4.61 |
| SB-FBSDE (Chen et al., 2022) | ✗ | ✗ | ✗ | 5.26 |
| VSDM (Deng et al., 2024) | ✗ | ✗ | ✗ | 4.24 |
| *Non-Memoryless SDE* | | | | |
| DSBM (Shi et al., 2023) | ✓ | ✗ | ✗ | 9.68 |
| **ASBM (Ours)** | ✓ | ✓ | ✓ | **3.16** |

*Table 1.* **FID evaluation on CIFAR-10.** ASBM shows superior generative performance by achieving all three key advantages: optimal coupling (OC), no reliance on pre-training (No PT), and consistent forward–backward dynamics (FB).

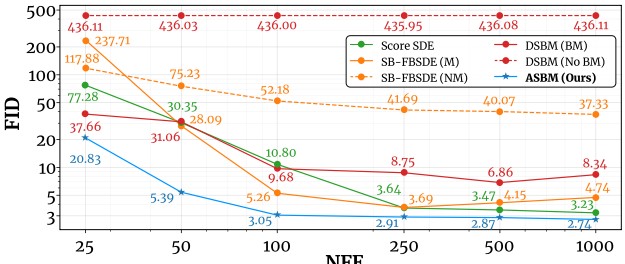

*Figure 3.* **FID comparison along the NFE.** We use M and NM to denote the memoryless and non-memoryless condition, respectively. BM denotes empirical bridge-matching pretraining.

| NFE | 25 | 50 | 100 | 250 | 500 | 1000 |
|---|---|---|---|---|---|---|
| Score SDE | 52.08 | 19.02 | 9.84 | 7.79 | 6.88 | 6.63 |
| **ASBM (Ours)** | **8.85** | **7.64** | **6.85** | **6.47** | **6.38** | **6.27** |

*Table 2.* FID evaluation on latent space of FFHQ.

**Initialization of $G^\psi$.** Following the score distillation (Yin et al., 2024b), we initialize $G^\psi$ from pretrained backward control $v_t^\phi$ via Tweedie's formula (Efron, 2011) at $t = 1$:

$$G^\psi(x) = \frac{x + (1 - \bar{\kappa}_1^2)v_1^\phi(x)/\sigma_1}{\bar{\kappa}_1}, \quad (19)$$

with notation as in Algorithm 1. For diffusion score models, this one-step estimate often produces noisy images due to its highly stochastic trajectory, typically mitigated by timestep shifting (Yin et al., 2023), which introduces bias. In contrast, ASBM's straighter path makes this initialization reliable without timestep shifting. See Appendix C for illustration.

**Advantages of ASBM Distillation.** ASBM not only yields a straighter path, but also constructs *highly organized trajectories* connecting *adjacent* pairs $(X_0, X_1)$. This is a special characteristic of non-memoryless SB originated from its *non-memoryless optimal coupling* (Proposition 3.1). Specifically, covering the entire prior space with non-memoryless trajectories leads to a more efficient connection from the specific mode in data space to the specific *nearby* area in $p_{\text{prior}}$. In other words, both $p_1^{u^\theta}(X_1 \mid X_0)$ and $p_0^{v^\phi}(X_0 \mid X_1)$ have lower variance than the case of memoryless process (See Sec. 4.2 for demonstrations). Together with straightness, it has strong benefits to distillation in the aspect of *learning difficulty* and *mode coverage* compared to the memoryless trajectory of diffusion models.

## 4. Experiments

We validate the generative performance and efficiency of ASBM in Sec. 4.1, and analyze its optimal trajectory in Sec. 4.2. Then, we verify effective distillation of ASBM

in Sec. 4.3. Finally, we explore the main hyperparameters through the ablation study in Sec. 4.4 and compare the training cost to score matching in Sec. 4.5.

**Dataset & Baselines.** We compare our ASBM against Score SDE (Song et al., 2021b) as a representative instantiation of memoryless diffusion and prior SB-based frameworks on the pixel space of CIFAR-10 (Krizhevsky et al., 2009) with FID (Heusel et al., 2017) metric. Then, we further verify ASBM in the LDM (Rombach et al., 2022) framework using the Stable Diffusion 3 autoencoder (Esser et al., 2024) on FFHQ (Karras et al., 2019). For distillation task, we report recall and precision metrics (Kynkäänniemi et al., 2019) to evaluate mode coverage, together with FID.

**Experimental Setup.** We adopt the Variance Preserving (VP) path (Song et al., 2021b) with non-memoryless setting as the base SDE for ASBM. For coupling generation at training, we use the Euler-Maruyama solver (Kloeden, 2011) with 20 NFE for CIFAR-10 and 50 NFE for FFHQ. More details can be found in Appendix E.

### 4.1. Image Generation Performance

**Generation on Pixel Space.** Tab. 1 reports FID on CIFAR-10 at 100 NFE using each method's reported solver. ASBM significantly outperforms all baselines by achieving the efficient and straight trajectory under optimal coupling. As discussed in Sec. 3.1, memoryless base SDEs (Score SDE, SB-FBSDE and VSDM) induce independent coupling at optimal state and thus fail to yield an informative optimal coupling. While DSBM relaxes this via a non-memoryless condition, it remains difficult to scale to high-dimensional data even with an additional pretraining stage.

We ablate SB-FBSDE with non-memoryless process and

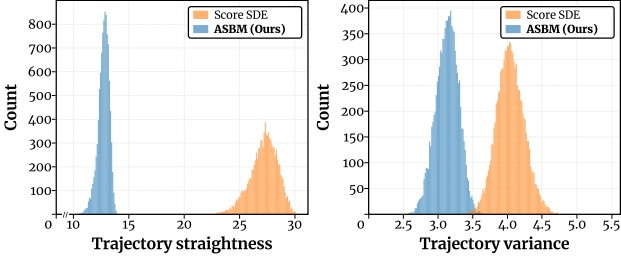

*Figure 4.* **Trajectory efficiency.** *Left*: Ours shows significantly straighter trajectory, leading to low NFE at generation. *Right*: Ours has lower trajectory variance, implying its better organized path.

| Method | Score SDE | SB-FBSDE | DSBM | **ASBM** |
|--------|-----------|----------|------|----------|
| FID    | 6.72      | 285.77   | 39.84 | **3.74** |

*Table 3.* FID at 25 steps with Heun solver on CIFAR-10.

DSBM without pretraining to evidently show these limitations in Fig. 3. (i) SB-FBSDE performs well only in the memoryless regime but substantially degrades under the non-memoryless setting, suggesting it cannot yield optimal coupling in high dimensions. (ii) DSBM works well only with empirical bridge matching pretraining, incorrectly assuming independent endpoint coupling as optimal one with the non-memoryless settings. Even with pretraining, DSBM shows unstable and inferior FID scores across various NFEs. In contrast, our ASBM achieves significantly lower FID with low NFE (20-100), implying its efficient trajectory.

**Generation in LDM Framework.** We verify generalizability of ASBM on latent space of FFHQ in Tab. 2. Consistent with the pixel-space results, ASBM achieves substantially lower FID at small NFE and maintains superior FID across the entire NFE range compared to Score SDE.

### 4.2. Optimal Trajectory of ASBM

We further analyze strength of our trajectory through path straightness/variance, and forward-backward consistency.

**Trajectory Straightness.** We measure the path straightness via the trajectory functional

$$S(X_{0:1}) := \frac{\sum_{i=0}^{T-1} \|X_{(i+1)/T} - X_{i/T}\|_2^2}{\|X_1 - X_0\|_2^2}, \quad (20)$$

from a $T$-step trajectory $\{X_{i/T}\}_{i=0}^T$. We note that in the fine-discretization limit, this metric is primarily governed by the diffusion magnitude rather than the geometric curvature of the drift. We use 10K trajectories generated with $T = 100$ steps. As shown in Fig. 4 (left), ASBM yields substantially smaller $S$ than Score SDE, confirming that our non-memoryless coupling produces straighter generation trajectories.

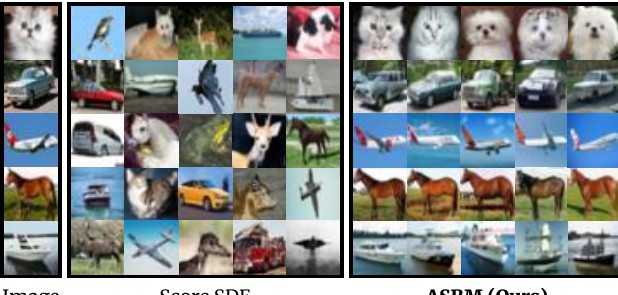

Image      Score SDE      **ASBM (Ours)**

*Figure 5.* **Localized prior-data coupling.** ASBM trajectories preserve information: reversing from a noised image produces samples similar to the original. In contrast, memoryless dynamics yield completely random samples due to highly noisy trajectories.

To complement this with a drift-aware measure, we additionally evaluate the drift directionality $\mathbb{E}_{t,X_t}[\cos(v_t(X_t), X_0 - X_1)]$, which captures how consistently the drift aligns with the overall noise-to-data direction. ASBM achieves $0.3672$ compared to $0.0602$ for Score SDE, indicating that the learned drift is substantially more directed toward the target.

**Trajectory Variance.** As discussed in Sec. 3.3, non-memoryless SB is expected to have *localized coupling*, which can be demonstrated by low trajectory variance where most mass of $p_0^{v^\phi}(X_0|X_1 = x_1)$ concentrates more on its centroid. To verify this property, for each of 10K initial noises, we generate 10 images and compute the average $\ell_2$ distance of these images to their centroid. As shown in Fig. 4 (right), ASBM exhibits a notably stronger concentration, indicating its strongly *organized* trajectory.

We verify this property by inversion test for further intuition. Starting from a fixed image $X_0$, we sample $X_1 \sim p^{u^\theta}(\cdot \mid X_0)$ using our forward SDE in Eq. (3), then reconstruct $\widehat{X}_0$ by running the backward SDE in Eq. (5) from $X_1$. As shown in Fig. 5, ASBM recovers $\widehat{X}_0$ that remains highly similar to original $X_0$, whereas Score SDE produces a random reconstruction, consistent with its memoryless dynamic. These results clearly indicate that ASBM's trajectory is not only straighter but also highly organized under non-memoryless process, which in turn simplifies distillation.

**Forward-Backward Consistency.** Finally, we evaluate the consistency of forward-backward dynamics. For an exact SB solution, the optimal bridge is unique, implying compatible forward-backward dynamics that share the same time marginals. We quantitatively evaluate this property through generation via the Heun's method (Ascher & Petzold, 1998; Karras et al., 2022), based on probability flow ODE (Song et al., 2021b) requiring precisely coupled forward-backward dynamics. Tab. 3 verifies the limitation of prior works by showing notable degradation of these baselines with Heun's method, whereas ASBM remains robust with a few steps, highlighting the benefit of our two-stage optimiza-

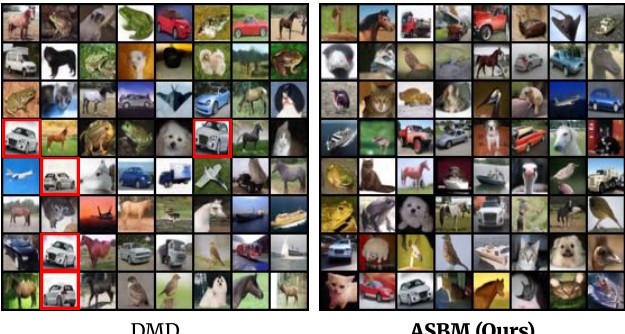

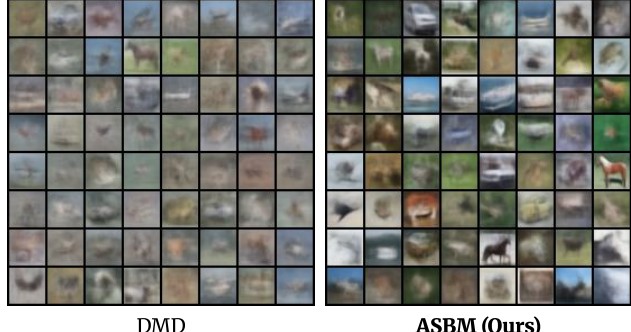

*Figure 6.* **Uncurated one-step generation from distillation on CIFAR-10**. Red boxes highlight repeated patterns indicating mode collapse. DMD still suffers from mode collapse despite the costly regression loss, whereas ASBM achieves diverse generation due to its organized, localized coupling.

| Method | FID ↓ | Recall ↑ | Precision ↑ |
|---|---|---|---|
| SDS (Poole et al., 2023) | 9.36 | 0.504 | 0.706 |
| DMD (Yin et al., 2024b) | 8.25 | 0.513 | **0.715** |
| **Ours** | **6.68** | **0.542** | 0.702 |

*Table 4.* Result of distillation to one-step generator on CIFAR-10.

tion. This is because, in practice, alternating optimization in SB-FBSDE and DSBM can be unstable in high dimensions and each direction provides inconsistent trajectory supervision to its counterpart optimization, eventually producing mismatched forward-backward systems.

### 4.3. Distillation to One-Step Generator

We showcase the ASBM's efficient trajectory via distillation to one-step generator (Sec. 3.3) on CIFAR-10, comparing it to score-based distillation baselines: Score Distillation Sampling (SDS) (Poole et al., 2023) and Diffusion Matching Distillation (DMD) (Yin et al., 2024b), which augments SDS with a regression loss to mitigate mode collapse.

**Performance.** As shown in Tab. 4, ASBM outperforms prior score-distillation models. Notably, the improved recall indicates substantially *reduced mode collapse* (see Fig. 6), avoiding the need of costly regression loss employed in DMD. This regression loss requires a large number of noise-image pairs (*e.g.*, 500K pairs) generated from the original score model. We attribute the superiority of ASBM's distillation to the localized prior–data coupling, *i.e.*, efficiently organized trajectory induced by our optimal coupling, which provides more informative guidance and better covers diverse data modes.

**Initialization.** Beyond generation quality, ASBM's localized prior–data coupling together with its straighter path provides a strong foundation for initializing the one-step generator. Standard diffusion-based distillation relies on

*Figure 7.* **Initialization of one-step generator on CIFAR-10.** Diffusion-based initialization (left) produces noisy images even with timestep shifting, while ASBM (right) yields clear initial estimates due to its straighter and more organized trajectory

Tweedie's one-step estimate, which produces highly noisy images due to the stochastic trajectory of diffusion models, requiring the theoretically inconsistent timestep shifting technique (Yin et al., 2024a) to mitigate the noise. In contrast, ASBM's straighter and more organized trajectory enables a direct application of Tweedie's formula (19) without any such correction, yielding a theoretically consistent initialization. As shown in Fig. 7, ASBM produces significantly clearer initial estimates compared to diffusion-based one, which in turn stabilizes training and leads to faster convergence with better performance and mode coverage.

### 4.4. Ablation Study

We investigate the degree of memorylessness and the effects of forward NFE on CIFAR-10.

**Memorylessness.** For the VP base process (Algorithm 1), a larger $\beta_{\max}$ injects more noise, and pushes the base SDE toward the memoryless process; for example, $\beta_{\max} = 20$ recovers the standard memoryless diffusion setting. We ablate memorylessness by varying $\beta_{\max}$. As shown in Fig. 8, $\beta_{\max}$ controls the trade-off between training efficiency and prior coverage. When $\beta_{\max}$ is too small (*e.g.*, $\beta_{\max} = 1$), the path becomes straighter and training is efficient, but the terminal distribution $p_1^{u^\theta}$ under-covers $p_{\text{prior}}$ even after successful optimization, because $p_1^{\text{base}}$ is too different from $p_{\text{prior}}$. This eventually leaves low-density holes in $p_1^{u^\theta}$, resulting in worse FID with fine-grained NFE. Conversely, a larger $\beta_{\max}$ (*e.g.*, $\beta_{\max} = 8$) improves coverage of the prior space, but the increased noise injection leads to more curved paths, making accurate simulation difficult with limited NFEs. We therefore use $\beta_{\max} = 4$ by default as a favorable trade-off between informative coupling and robust prior matching.

**Forward NFE.** Forward NFE largely determines the training cost. Prior SB methods (Chen et al., 2022; Shi et al., 2023) typically require 100–200 NFEs to alternately simulate forward-backward dynamics, which dominates run-

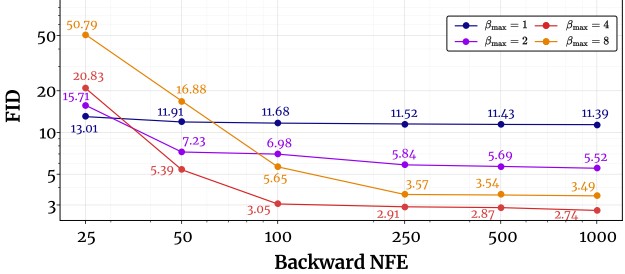

*Figure 8.* Ablation on different degree of memorylessness.

| Forward | Backward NFE | | | | | |
|---|---|---|---|---|---|---|
| NFE | 25 | 50 | 100 | 250 | 500 | 1000 |
| 10 | 23.62 | 8.03 | 3.58 | 3.40 | 3.28 | 3.05 |
| 20 | 20.83 | **5.39** | **3.05** | **2.91** | **2.87** | **2.74** |
| 50 | **20.73** | 5.87 | 3.16 | 3.01 | 2.94 | 2.77 |

*Table 5.* Ablation on different forward NFE.

time, while ASBM only uses lightweight forward simulation, achieving strong performance with just 20 NFEs. As shown in Tab. 5, ASBM remains strong even with 10 NFEs, and the negligible gap between 20 and 50 NFEs supports our hypothesis that the data-to-energy forward optimization is easier, enabling accurate couplings with fewer NFEs.

### 4.5. Training Efficiency

Our backward optimization converges significantly faster, *i.e.*, 600 epochs for ASBM and 3300 epochs for Score SDE (Song et al., 2021b), due to its supervision under *optimal coupling*. As discussed in Sec. 3.2, our forward dynamic is also lightweight due to its data-to-energy direction, converging with the cost same as 150 backward epochs. Considering the cost of coupling construction, the total training cost of ASBM is equivalent to 2100 Score SDE epochs, yielding a $0.64\times$ reduced computation relative to Score SDE. In terms of wall-clock time on a single A100 40GB GPU, ASBM takes approximately 100 hours for CIFAR-10, compared to 144 hours for Score SDE.

## 5. Related Work

**Generative Models.** Dynamic-based generative models have become a major paradigm in generative tasks (Ho et al., 2020; Song et al., 2021a;b; Lipman et al., 2023; Liu et al., 2023b; Albergo et al., 2025; De Bortoli et al., 2021). In order to improve the efficiency of these models, research efforts devoted to reducing the NFE of these models, *e.g.*, different dynamic solvers (Lu et al., 2022; 2025; Zhang & Chen, 2023; Karras et al., 2022), or rectifying the trajectories (Liu et al., 2023b; 2024b). However, these works are based on post-refinement on the top of the noisy diffusion path. To obtain the optimal trajectory, study on dynamic optimal transport has been urged.

**Dynamic Optimal Transport (OT).** OT-based generative models range from Wasserstein objectives (Arjovsky et al., 2017) to entropy-regularized OT, *i.e.*, Schrödinger Bridges (SB) (Léonard, 2014; Chen et al., 2016; 2021; Caluya & Halder, 2021; Vargas et al., 2021; Tong et al., 2024b). Scalable SB solvers typically alternate forward-backward

updates, *e.g.*, Iterative Proportional Fitting/Sinkhorn-style methods (Fortet, 1940; Kullback, 1968; Rüschendorf, 1995; De Bortoli et al., 2021; Chen et al., 2022) and recent matching-based variants improving practicality (Shi et al., 2023; Chen et al., 2023; Peluchetti, 2023). SB has been applied to image generation (De Bortoli et al., 2021; Chen et al., 2022; Deng et al., 2024) and image-to-image translation (Liu et al., 2023a; Theodoropoulos et al., 2024; Gushchin et al., 2024). Our approach is fundamentally different, as we first formulate generative modeling as finding optimal coupling and then supervise the generation path under this coupling to learn efficient trajectory.

**Stochastic Optimal Control (SOC).** SOC-based sampling methods learn a control that steers a base diffusion process toward a target Boltzmann distribution (Zhang & Chen, 2022; Vargas et al., 2023; Havens et al., 2025; Liu et al., 2025). Among these, adjoint matching (Domingo-Enrich et al., 2025; Shin et al., 2026)-based sampling methods (Havens et al., 2025; Liu et al., 2025) have shown strong performance, and ASBS (Liu et al., 2025) further extends this line by supporting non-memoryless base dynamics. However, these methods target sampling from unnormalized densities, not generative modeling from data. We bridge this gap by casting the SB forward dynamic as a data-to-energy sampling problem, leveraging SOC to construct optimal couplings for solving SB problem.

## 6. Conclusion

We present ASBM, a two-stage SB framework that learns *informative* optimal couplings. The key idea is to decouple forward and backward optimization: we first learn the forward dynamic as a controlled sampling problem under a *non-memoryless* base process, then train the backward dynamic using the optimal couplings induced by the learned forward transport. This avoids unstable bidirectional alternating training, yielding an efficient trajectory. While our extensive experiments thoroughly validate ASBM on standard benchmarks (pixel space, LDM framework, distillation), we leave large-scale high-resolution and conditional generation to future work, expecting the natural transfer. Moreover, since our framework accommodates any energy-defined prior, exploring alternative priors better suited to specific data modalities is a promising direction.

## Acknowledgments

This work was also supported by Samsung Electronics, Youlchon Foundation, National Research Foundation of Korea (NRF) grants (RS-2021-NR05515, RS-2024-00336576, RS-2023-0022663, RS-2025-25402648, RS-2024-00349646, RS-2024-00342044), and the Institute for Information & Communication Technology Planning & Evaluation (IITP) grants (RS-2022-II220264, RS-2024-00353131) funded by the Korean government.

## Impact Statement

Our work focuses on developing computational methods for image generation with optimal trajectory. The techniques are purely theoretical and computational, relying exclusively on public image dataset. No personal data, or sensitive contents are involved. We therefore identify no ethical concerns arising from this research. For this framework, there are many possible societal impacts, none of which need specific highlighting.

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

# Appendix

## A. Proofs

**Stochastic Optimal Control (SOC).** In our framework (in general, SOC transports from samplable distribution to Boltzmann distribution), SOC-based sampling problem finds a control that transports $p_{\text{data}}$ to a prescribed prior $p_{\text{prior}} \sim \exp(-E(x))$ under cost minimization:

$$\min_u \ \mathbb{E}_{X \sim p^u} \left[ \int_0^1 \frac{1}{2} \|u_t(X_t)\|^2 \, \mathrm{d}t + g(X_1) \right] \tag{21}$$

$$\text{s.t. } \mathrm{d}X_t = \left[ f_t(X_t) + \sigma_t u_t(X_t) \right] \mathrm{d}t + \sigma_t \, \mathrm{d}W_t, \quad X_0 \sim p_{\text{data}}, \tag{22}$$

where $g(x) : \mathbb{R}^d \to \mathbb{R}$ is a terminal cost and SDE is constrained only by data distribution $X_0 \sim p_{\text{data}}$. Note that the terminal constraint on $p_{\text{prior}}$ is implicitly enforced by the terminal cost $g(x)$.

Under the SOC optimality, optimal control can be analytically derived through Hamilton-Jacobi-Bellman (HJB) equation (Bellman, 1954).

**Theorem A.1.** *(SOC optimality) Under the SOC optimality, the optimal control is $\overrightarrow{u}_t^\star(x) = -\sigma_t \nabla V_t(x)$, where $V_t(x) : [0,1] \times \mathbb{R}^d \to \mathbb{R}^d$ is a value function,*

$$V_t(x) = -\log \mathbb{E}_{X \sim p^{\text{base}}}[\exp(-g(X_1)) \mid X_t = x]. \tag{23}$$

*The optimal joint distribution can also be characterized as*

$$p^\star(X_0, X_1) = p^{\text{base}}(X_0, X_1) \exp(-g(X_1) + V_0(X_0)). \tag{24}$$

**Proof of Proposition 3.1.**

*Proof.* Consider the memoryless condition

$$p_{0,1}^{\text{base}}(X_0, X_1) \stackrel{\text{memoryless}}{:=} p_0^{\text{base}}(X_0) \, p_1^{\text{base}}(X_1). \tag{25}$$

Under the memoryless condition (8), the initial value function (23) becomes:

$$V_0(X_0) \stackrel{\text{memoryless}}{=} -\log \int p^{\text{base}}(X_1) \exp(-g(X_1)) \, \mathrm{d}X_1. \tag{26}$$

Note that right-hand side is constant to $X_0$. Substituting (26) and (25) into (24) yields the factorization

$$p^\star(X_0, X_1) = p^{\text{base}}(X_0) \frac{p^{\text{base}}(X_1) \exp(-g(X_1))}{\int p^{\text{base}}(X_1) \exp(-g(X_1)) \, \mathrm{d}X_1}. \tag{27}$$

As a result, $X_0$ and $X_1$ are independent under $p^\star$, directly indicating that *non-independent* optimal couplings cannot be recovered under the memoryless condition. $\square$

## B. Terminal Cost in SOC-based Sampling Problem

**Case of Memoryless Base SDE.** To remove the bias introduced by $V_0(X_0)$ in Eq. (24), a common approach is the adoption of memoryless condition (25). Starting from Eq. (24),

$$p^*(X_1) = \int p^*(X_0, X_1) \, \mathrm{d}X_0 = \int p^{\text{base}}(X_0, X_1) \exp(-g(X_1) + V_0(X_0)) dX_0 \tag{28}$$

$$= \int p^{\text{base}}(X_0) \, p^{\text{base}}(X_1) \exp(-g(X_1) + V_0(X_0)) \, \mathrm{d}X_0 \tag{29}$$

$$\propto p^{\text{base}}(X_1) \exp(-g(X_1)) = p_{\text{prior}}(X_1), \tag{30}$$

where second equality holds for memoryless condition (25). As a result, we can set terminal cost as

$$g(x) = \log \frac{p_1^{\text{base}}(x)}{p_{\text{prior}}(x)} \tag{31}$$

for SOC-based sampling problem with memoryless base SDE (Zhang & Chen, 2022; Peluchetti, 2023; Havens et al., 2025).

**Case of Non-Memoryless Base SDE.** ASBS (Liu et al., 2025) generalize the sampling problem to non-memoryless dynamic. To resolve the bias by initial value function $V_0(X_0)$ in Eq. (28) without reliance on memoryless condition Eq. (25), they further deploy the SB optimality.

**Theorem B.1.** *(Optimal control in SB problem) Under the SB optimality (Pavon & Wakolbinger, 1991; Chen et al., 2021; Caluya & Halder, 2021), the optimal control is*

$$u_t^\star(x) = \sigma_t \nabla_x \log \varphi_t(x), \quad v_t^\star(x) = \sigma_t \nabla_x \log \hat{\varphi}_t(x) \tag{32}$$

*where $\varphi_t, \hat{\varphi}_t \in C^{1,2}([0,1], \mathbb{R}^d)$ are SB potentials satisfying*

$$\varphi_t(x) = \int p_{1|t}^{\text{base}}(y \mid x)\, \varphi_1(y)\, dy, \quad \varphi_0(x)\, \hat{\varphi}_0(x) = p_{\text{prior}}(x),$$

$$\hat{\varphi}_t(x) = \int p_{t|0}^{\text{base}}(x \mid y)\, \hat{\varphi}_0(y)\, dy, \quad \varphi_1(x)\, \hat{\varphi}_1(x) = p_{\text{data}}(x).$$

Under the SOC optimality Theorem A.1 and SB optimality Theorem B.1, we can obtain the transform

$$\varphi_t(x) = \exp(-V_t(x)), \quad \hat{\varphi}_t(x) = \exp(V_t(x))\, p_t^\star(x), \tag{33}$$

which connects between SOC value function and SB potentials. This leads to setting terminal cost as

$$g(x) = \log \frac{\hat{\varphi}_1(x)}{p_{\text{prior}}(x)}. \tag{34}$$

Applying Adjoint Matching (AM) (Domingo-Enrich et al., 2025) to SOC objective (21) with this terminal cost (34) under VP base SDE yields

$$u^* = \arg\min_u \mathbb{E}_{p^{\bar{u}}} \left[ \| u_t(X_t) + \kappa_t \sigma_t (\nabla E(X_1) + \nabla \log \hat{\varphi}_1(X_1)) \|^2 \right], \tag{35}$$

where we use the notation in Algorithm 1. However, we need an access to $v \nabla \log \hat{\varphi}_1(x))$ to make this objective feasible. To resolve this problem, Corrector Matching (CM) is introduced, which can be derived by variational form of $\nabla \log \hat{\varphi}_1(x)$:

$$\nabla \log \hat{\varphi}_1 = \arg\min_h \mathbb{E}_{p_{0,1}^*} \left[ \| h(X_1) - \nabla_{x_1} \log p^{\text{base}}(X_1|X_0) \|^2 \right]. \tag{36}$$

Finally, following the adoption of reciprocal projection (Havens et al., 2025), we can alternately train these two objectives with parameterized models as

$$\min_\theta \; \mathbb{E}_{p_{t|0,1}^{\text{base}},\, p_{0,1}^{\bar{u}\theta}} \left[ \left\| u_t^\theta(X_t) + \left( \sigma_t \nabla E + \bar{v}_1^\phi \right)(X_1) \right\|^2 \right], \tag{37}$$

$$\min_\phi \; \mathbb{E}_{p_{0,1}^{\bar{u}\theta}} \left[ \left\| v_1^\phi(X_1) - \sigma_1 \nabla_{x_1} \log p^{\text{base}}(X_1 \mid X_0) \right\|^2 \right]. \tag{38}$$

## C. Distillation

**Derivation for SB Distillation Framework.** Upon successful training, we are given the pretrained backward dynamic,

$$dX_t = \left[ f_t(X_t) - \sigma_t v_t^\phi(X_t) \right] dt + \sigma_t dW_t, \quad X_1 \sim p_{\text{prior}}, \tag{39}$$

which approximately transports prior distribution to data distribution $p_{\text{data}}$. Denote the path measure induced by backward dynamic (39) as $p^\phi$.

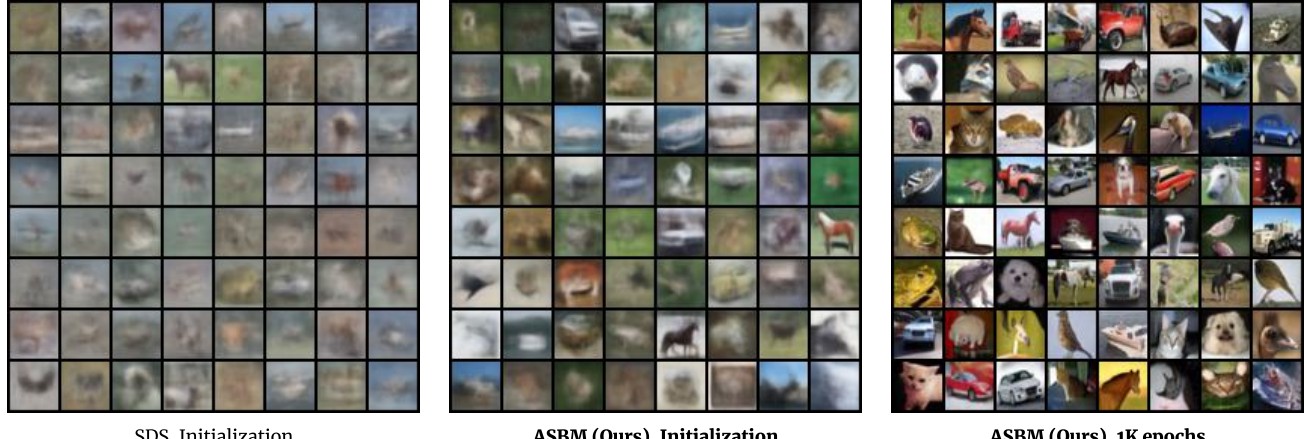

SDS, Initialization          **ASBM (Ours), Initialization**          **ASBM (Ours), 1K epochs**

*Figure I.* **Initialization of one-step generator (Uncurated generation).** As discussed in Sec. 3.3, ASBM shows clear initialization for one-step generator, indicating its straighter and more organized trajectory. On the other hand, score-based initialization gives much more noisy initialization even with timestep shifting (Yin et al., 2024b).

Now consider the one-step generator $G^\psi : \mathbb{R}^d \times \mathbb{R}^d \to \mathbb{R}^d$, which defines the output distribution,

$$p_0^\psi := \mathrm{Law}\big(G^\psi(X_1, z)\big), \tag{40}$$

where $X_1 \sim p_{\mathrm{prior}}$, $z \sim \mathcal{N}(0, I)$. Although we define the one-step generator as $G^\psi : \mathbb{R}^d \to \mathbb{R}^d$ which only takes $X_1$ in main paper for simplicity, it is more general to define it as Eq. (40) to consider the stochasticity via noise $z$.

The most straightforward way is to minimize the KL divergence between $p_0^\phi$ and $p_0^\psi$, which is infeasible. Instead, we distill our learned backward control $v_t^\phi$ in continuous-time control space.

Assume an optimal backward control $v^\xi : [0, 1] \times \mathbb{R}^d \to \mathbb{R}^d$ such that its corresponding controlled SDE

$$\mathrm{d}X_t = \big[(f_t(X_t) - \sigma_t v_t^\xi(X_t)]\big)\,\mathrm{d}t + \sigma_t\,\mathrm{d}W_t, \;\; X_1 \sim p_{\mathrm{prior}}, \tag{41}$$

induces the terminal marginal $X_0 \sim p_0^\xi \equiv p_0^\psi$. Let $p^\xi$ denote its path measure. Then, by data processing inequality, KL divergence between terminal distributions is bounded by

$$D_{\mathrm{KL}}\big(p_0^\xi \,\|\, p_0^\phi\big) \;\leq\; D_{\mathrm{KL}}\big(p^\xi \,\|\, p^\phi\big). \tag{42}$$

Girsanov's theorem (Särkkä & Solin, 2019) turns the path-space KL divergence (42) into a tractable drift-matching loss that can be estimated from sampled trajectories:

$$\frac{1}{2}\,\mathbb{E}_{p^\xi}\left[\int_0^1 \left\|v_t^\xi(X_t) - v_t^\phi(X_t)\right\|^2 dt\right]. \tag{43}$$

Since we assume $v_t^\xi$ to be optimal, $p^\xi$ follows reciprocal process, leading to,

$$\min_\psi \; \mathbb{E}_{p_{t|0,1}^{\mathrm{base}}, X_0, X_1}\big[\big\|\bar{v}_t^\xi(X_t) - \bar{v}_t^\phi(X_t)\big\|^2\big], \tag{44}$$

where $X_0 \sim G^\psi(X_1, z)$, $X_1 \sim p_{\mathrm{prior}}$.

Then, inspired by the distillation frameworks in DMs (Poole et al., 2023; Yin et al., 2024b), we initialize $v_t^\xi$ from the pretrained $v_t^\phi$ and dynamically update it with bridge matching loss to reflect the continuously changing $p_0^\psi$:

$$\min_\xi \; \mathbb{E}_{p_{t|0,1}^{\mathrm{base}}, X_0, X_1}\big[\big\|v_t^\xi(X_t) - \sigma_t \nabla_{x_t} \log p^{\mathrm{base}}(X_t \mid X_0)\big\|^2\big], \tag{45}$$

where $X_0 \sim G^\psi(X_1, z)$, $X_1 \sim p_{\mathrm{prior}}$. As a result, we mainly optimize $G^\psi$ via Eq. (44) while dynamically updating $v_t^\xi$ via Eq. (45).

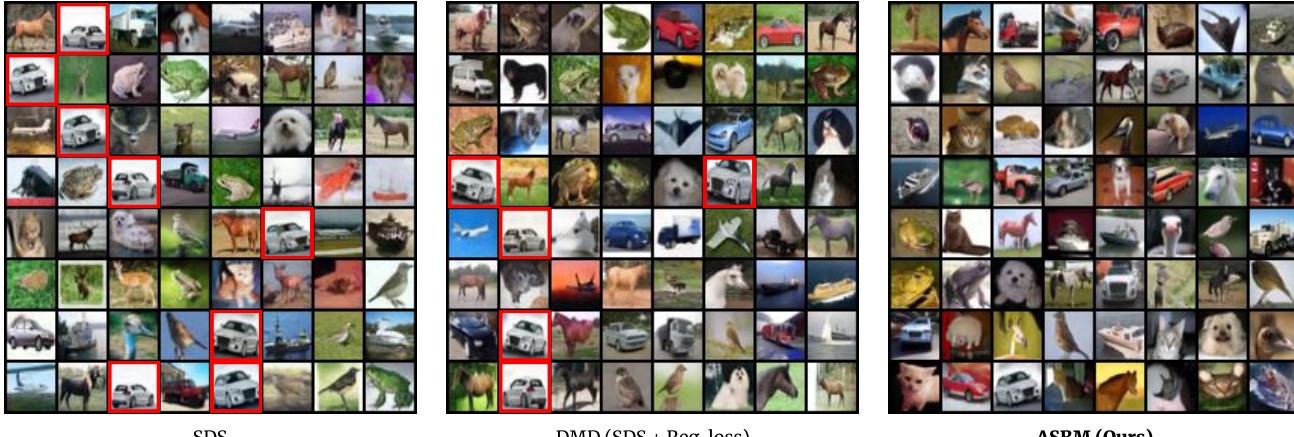

| SDS | DMD (SDS + Reg. loss) | **ASBM (Ours)** |

*Figure II.* **Mode collapse in distillation (Uncurated generation).** ASBM shows strong mode coverage in distillation task while the score distillation models still suffer from mode collapse even with costly regression loss.

With memoryless condition (25), reciprocal process in Eq. (44) and Eq. (45) collapses to conditional path in standard diffusion model, and our distillation framework becomes exactly same as score distillation framework (Poole et al., 2023; Yin et al., 2024b).

**Initialization for One-Step Generator.** As discussed in Sec. 3.3, ASBM provides strong initialization for one-step generator $G^\psi$ with Tweedie's formula (19) due to its significantly straighter and efficiently organized trajectory. As shown in Fig. I, ASBM yields clearly better one-step estimate than score-based initialization, which makes the distillation easier than that from memoryless diffusion.

**Better Mode Coverage of ASBM.** Score-distillation models suffer from mode collapse problem (Yin et al., 2024b) which inevitably requires further refinement through regression loss (Yin et al., 2024b) or adversarial loss (Yin et al., 2024a). Especially, regression loss requires generation of huge amount of noise-image pairs from original score model which is highly costly. As shown in Fig. II, pure score distillation (SDS) (Poole et al., 2023) exhibits severe mode collapse. Even with an additional heavy regression loss, DMD (Yin et al., 2024b) can still collapse. In contrast, ASBM substantially mitigates mode collapse, which we attribute to its organized and efficient trajectories, as demonstrated in Sec. 4.3.

## D. Empirical Validation of Prior Matching

We verify that the learned forward process in ASBM accurately transports the data distribution to the target prior $p_{\text{prior}} = \mathcal{N}(0, I)$. Specifically, we evaluate the terminal samples $X_1 \sim p_1^{u_\theta}$ obtained by simulating the forward controlled SDE (3) from $X_0 \sim p_{\text{data}}$ on CIFAR-10 ($D = 3072$, $N = 50\text{K}$ samples).

We consider three complementary diagnostics, each targeting a different aspect of Gaussianity:

1. **Scalar-level mean and variance.** For a standard Gaussian, the per-coordinate mean and variance should be close to $(0, 1)$.

2. **Squared norm statistics.** The mean and variance of $\|X_1\|_2^2$ should be close to $(D, 2D)$, since $\|X_1\|_2^2 \sim \chi^2(D)$ under the true prior.

3. **Covariance spectrum.** The minimum and maximum eigenvalues of the sample covariance matrix should concentrate near $\left(1 - \sqrt{D/N}\right)^2$ and $\left(1 + \sqrt{D/N}\right)^2$, respectively.

Tab. I reports the results. The learned forward process closely matches the Gaussian reference across all three diagnostics, confirming that the terminal distribution $p_1^{u_\theta}$ provides adequate coverage of $p_{\text{prior}}$.

*Table I.* Prior alignment diagnostics on CIFAR-10. Each metric compares $N{=}$50K forward-simulated samples against true Gaussian samples of the same size ($D{=}$3072).

|  | Scalar (mean, var) | $\|X_1\|_2^2$ (mean, var) | Eigenvalues (min, max) |
| --- | --- | --- | --- |
| Gaussian reference | $(0, 1)$ | $(3072, 6144)$ | $(0.566, 1.557)$ |
| ASBM (learned forward) | $(-0.001, 1.012)$ | $(3090, 6322)$ | $(0.572, 1.594)$ |

## E. Experiment Settings

**Model Architecture.** We use UNet (Ronneberger et al., 2015) architecture following the hyperparameters in (Tong et al., 2024a). For backward dynamic, we use 4 residual blocks for each channel following the Score SDE (Song et al., 2021b) and 2 residual blocks for our forward dynamic which significantly reduces the computation cost. For LDM experiment, we employ Stable Diffusion 3 (Esser et al., 2024) autoencoder. We set batch size as 128.

**Training Environment.** We conduct all the experiments with a single NVIDIA A100 40 GB.

**Training Time.** It takes about 4 days for ASBM (2100 epochs) and 6 days for Score SDE (3300 epochs) on single A100.

