# OpenReview forum: "Efficient Generative Modeling beyond Memoryless Diffusion via Adjoint Schrödinger Bridge Matching"
_ICML.cc/2026/Conference — ICML 2026 regular_

### Official Review · Reviewer_Lx1Q · 2026-03-05

**Soundness:** 3
**Presentation:** 3
**Significance:** 2
**Originality:** 2
**Overall Recommendation:** 5
**Confidence:** 3

**Summary:**

This paper, ASBM, proposes a generative modeling framework  based on the Schrödinger Bridge (SB) formulation by using a non-memoryless base SDE. This framework consists of decoupling the optimization of implicit forward and backward dynamics into two stages. In Stage 1, the forward control is formulated as a Stochastic Optimal Control (SOC), allowing the model to learn an optimal trajectory (coupling) between the endpoint. In Stage 2, the backward control is trained using Bridge Matching under the learned coupling. The authors argue that this optimization improves stability and learns straighter trajectory, enabling efficient sampling with fewer NFEs. In addition, this paper proposes one-step distillation method based on ASBM framework. The paper demonstrates competitive performance on image generation.

**Compliance With Llm Reviewing Policy:**

Affirmed.

**Final Justification:**

The rebuttal resolved my concerns regarding prior alignment, methodological novelty, and computational efficiency, leading to a score increase.

Specifically, the empirical results confirmed that the learned forward process accurately reaches the target prior. The clarification of the two-stage optimization strategy successfully distinguished ASBM from existing SB methods.

**Key Questions For Authors:**

Q. The method requires sampling $X_1$ using the learned forward control $u_t$ and backward control $v_t$. What is the additional computational overhead introduced by sampling $X_1$? Reporting training or sampling time/NFE relative to baselines would help clarify the practical efficiency.

**Limitations:**

Yes.

**Strengths And Weaknesses:**

Strengths:

**1. Efficient generation sampling.**

This paper shows the ability to generate high-quality samples with lower NFEs, including one-step distillation. The results suggest that ASBM can maintain good fidelity even with a small number of function evaluations (Fig 3, Table 2).

**2. Clear demonstration of straight trajectory.**

Fig 4 and Table 3 provide that the learned trajectories are straighter. This supports that the proposed formulation can learn more efficient transport paths between the data distribution and the prior.

**3. Comprehensive experiments.**

The experiments are thorough and well organized. The experimental comparisons with prior SB methods are reasonable, and the ablation studies provide useful insights into the behavior of the model.


Weaknesses:

**1. Prior alignment validation.**

The paper claims that the base SDE injects smaller noise compared to the forward process in standard diffusion models (L.188), which is one of the reasons why the learned trajectories become straighter. Could the authors provide empirical evidence showing that the learned forward process produces samples that match the prior distribution? This seems especially relevant in light of the ablation study in Figure 6, which suggests that insufficient noise injection may degrade performance.

**2. Novelty relative to prior work ASBS [1].**

Equations (12) and (13) appear to be central to the training procedure of ASBM, but these objectives seem closely related to those introduced in prior work on Adjoint Schrödinger Bridge Sampler (ASBS) [1]. From my understanding, ASBS alternates between forward and backward updates using adjoint matching and corrector matching, whereas ASBM performs a decoupled optimization: the forward control is first optimized (while only learning the corrector at $t=1$), and the full backward dynamics are learned afterward using the learned forward coupling. The paper could clarify more explicitly what the decisive difference is between ASBS and the proposed ASBM framework, and why the proposed two-stage optimization leads to improved stability or performance.

**3. Generality of the energy formulation.**

In the paper, does the energy function $E(x)$ correspond to the Gaussian prior? Would the proposed framework also work with other energy functions (e.g., Laplacian or other priors), or does the method rely on properties specific to Gaussian distributions?

[1] Liu, G.-H., Choi, J., Chen, Y., Miller, B. K., and Chen, R. T. Adjoint schrödinger bridge sampler. arXiv preprint arXiv:2506.22565, 2025.

---

> ### Author Rebuttal · Authors · 2026-03-31
>
> We deeply appreciate the reviewer's careful reading and valuable suggestions regarding the clarifications for the important details of our work.
>
> **W1. Empirical evidence of covering marginal prior distribution**
>
> We agree that validating whether the learned forward process actually reaches the prior marginal is both important in itself and closely related to the memorylessness ablation in Fig. 6. For this reason, we provide 3 evaluation metrics and the results.
>
> Specifically, we evaluated with $N$ forward-simulated samples ($D$ dimension). We considered three complementary diagnostics:
> (1) the "scalar-level mean and variance", which for a standard Gaussian should be close to $(0,1)$;
> (2) the "mean and variance of the squared vector norm", which should be close to $(D, 2D)$; and
> (3) the "minimum and maximum eigenvalues of the covariance matrix", which measure isotropy and should be close to $\left(1-\sqrt{D/N}\right)^2$ and $\left(1+\sqrt{D/N}\right)^2$, respectively.
>
> On CIFAR-10, with $D=3072$ and $N=50\text{K}$, true Gaussian samples yield the reference values $(0,1)$, $(3072, 6144)$, and $(0.566, 1.557)$ for these three metrics. The learned forward process in ASBM produces $(-0.001, 1.012)$, $(3090, 6322)$, and $(0.572, 1.594)$, respectively, which are very close to the Gaussian reference.
>
> We hope this provides empirical evidence that the learned forward process matches the target prior distribution well in practice. We will include it in the main paper.
>
> **W2.1. Difference to ASBS**
>
> The key difference is in the problem setting and scope. ASBS is an SOC-based method for sampling from an energy-defined, unnormalized target distribution, i.e., it learns only the forward dynamics from an empirical distribution to an energy-specified distribution. In contrast, our goal is to solve the SB problem for prior-to-data generation, which requires learning both forward and backward dynamics.
>
> Concretely, our method first uses the ASBS to learn the easier data-to-prior forward dynamics, and then uses the learned forward optimal coupling to supervise the backward dynamics for generation. In this sense, ASBS addresses only one direction, whereas ASBM is designed to solve the full generative SB problem by additionally learning the coupled backward process.
>
> **W2.2. Reasons for performance gains in two-stage optimization**
>
> Existing SB methods typically optimize forward and backward dynamics simultaneously, so each direction is supervised using imperfect trajectories generated by the other direction dynamic, which makes training inherently unstable. This issue becomes especially severe for non-memoryless schedules due to high instability. In contrast, ASBM first learns the forward process alone, which is much more stable to learn because it maps complex data to a simple prior. Once this forward process is learned, it provides a well-organized coupling that enables much more stable supervision of the backward dynamics. We believe this is what enables stable and efficient learning of non-memoryless SBs in practice, and ultimately leads to better generation performance through straighter and more organized trajectories.
>
> **W3. Can ASBM work with other energy functions (prior distribution)?**
>
> In our current experiments, the energy function $E(x)$ corresponds to the energy of Gaussian prior. However, the proposed framework is not restricted to Gaussian energies. In principle, it can be applied to any prior distribution specified in energy form.
>
> We therefore do not rely on properties unique to the Gaussian distribution, beyond its convenience and standard use in image generation. We believe that exploring alternative energy functions or priors that are better suited to image generation is an interesting direction for future work, and we will clarify this point in the paper.
>
> **Q1. Cost of forward simulation to get data-prior coupling during training**
>
> Forward simulation accounts for about 60% of the total training time, which makes efficient forward inference particularly important. In ASBM, since forward dynamic goes from complex data to a simple prior, it allows accurate simulation with very few NFEs (20, and even 10 shows comparable performance in Table. 5), resulting in 0.65X reduced total training time for convergence compared to Score SDE. By contrast, existing SB methods typically require alternating forward and backward simulations, both of which are highly costly often requiring 100–200 NFE.
>
> For test-time sampling, the cost is identical as long as base model architecture is same. In particular, ASBM and Score SDE have exactly the same sampling cost in our experiments.

---

> > ### Author Rebuttal · Reviewer_Lx1Q · 2026-04-01
> >
> > I appreciate the authors' detailed rebuttal. The responses have successfully addressed my initial concerns. Since my theoretical and empirical questions have been adequately resolved, I will raise my score accordingly.

---

> > > ### Author Response · Authors · 2026-04-01
> > >
> > > We deeply appreciate your highly constructive feedback throughout the review process, as well as your decision to raise the score. We are glad to hear that our rebuttal has successfully resolved your concerns. We will make sure to carefully include all the discussed updates and clarifications in the final manuscript.

---

### Official Review · Reviewer_EjWW · 2026-03-10

**Soundness:** 3
**Presentation:** 3
**Significance:** 3
**Originality:** 3
**Overall Recommendation:** 4
**Confidence:** 4

**Summary:**

This paper proposes ASBM, a novel generative modeling framework that decomposes the Schrödinger Bridge problem into two stages: (1) learning an optimal forward control via stochastic optimal control to construct informative endpoint couplings, and (2) training a backward control via bridge matching using these couplings.

**Compliance With Llm Reviewing Policy:**

Affirmed.

**Final Justification:**

it has effectively addressed some of my concerns. Regarding W1, W2, and W5, I believe the authors have answered my questions well, and overall I do not have many remaining issues with experimental setup and methodology. However, I still have some follow-up questions regarding your theoretical sections.

**W3. On the definition of straightness:** the paper defines straightness as $S=\frac{\mathbb{E}\sum_i^{T-1}|X_{i+1/T}-X_{i/T}|^2}{\mathbb{E}|X_0-X_1|^2}$. Considering only the numerator, as $T \to \infty$, for any SDE path this becomes the quadratic variation $\int_0^1 \mathbb E|dX_t|^2 = \int_0^1 \sigma_t^2 dt$. Consequently, it depends only on the diffusion term and is independent of the drift term. I believe this comparison is not reasonable, as it only ensures that the diffusion term becomes smaller, without reflecting the role of the drift term. Therefore, only your first explanation holds  that the "straightness" as you defined it decreases because the diffusion term correspondingly decreases. I think it is very difficult to define straightness for SDEs using Euclidean metrics. Therefore, I would suggest that the authors clarify that their definition of straightness primarily reflects the diffusion magnitude rather than the geometric curvature of paths. Alternatively, if the authors wish to claim 'straighter paths,' they may need to provide a definition that accounts for both drift and diffusion components.

**W4. On the definition of optimal transport coupling:** It seems that it is some form of energy minimization over two stochastic process paths. However, the standard optimal transport problem considers the minimization of some cost defined between two distributions. I think this point requires further clarification. So I maintain my recommendation.

**Key Questions For Authors:**

1. About experiments: Algorithm 1 requires additional training for the forward/backward control parameters. Could you provide a comparison of the training cost of the proposed method with the other methods in Table 1(likely pure diffusion process, SiT, DiT or Score based model)?

2. Moreover, the experiments appear to be validated only on CIFAR-10. Although the theory is well-developed, the results lack persuasiveness to some extent. Could you provide experiments on more diverse and higher-resolution datasets?

3. About straightness: This paper claims that "ASBM produces significantly straighter and more efficient sampling paths. "  While the empirical results (Fig 3) show that ASBM trajectories appear straighter than diffusion models, the paper lacks theoretical justification for this observation. It would strengthen the paper to discuss why ASBM yields straighter paths.

4. About optimal map: This paper claims that "Our approach addresses the fundamental inefficiencies of diffusion models, i.e., curved trajectories and noisy training targets, by explicitly recovering optimal transport couplings." However, it never defines what optimal transport cost is being minimized. For the VP SDE base process, what is the corresponding cost function? Without a clear definition, the claim of 'optimal transport' is ambiguous.

5.  About the base SDE: It seems that this base SDE can only choose the SDE that diffusion term is not zero. We can't choose probability flow(diffusion term=0) as our basic SDE, for Eq. 13 may explode. Could the method be extended to allow deterministic base processes, perhaps by using a different matching objective?

**Limitations:**

see Key questions

**Strengths And Weaknesses:**

Strengths: ASBM offers a principled, efficient, and empirically superior approach to generative modeling, with clear advantages in sample quality, sampling speed, and distillation performance.

Weakness: See Key Questions

---

> ### Author Rebuttal · Authors · 2026-03-31
>
> We are grateful for the reviewer's thoughtful comments, notably the requests for clarifications on our contributions and extra experiments. Given the strict character constraints, we respectfully direct you to our replies to the other reviewers for answers to overlapping questions, indicated in each comment.
>
> **W1. Training cost comparison to pure Score SDE**
>
> The overall training cost of ASBM remains significantly lower than Score SDE. As reported in Sec. 4.5 and Appendix. D, ASBM requires only about 0.64X the training cost of Score SDE. We attribute this training efficiency mainly to two factors: (i) the very low simulation cost of the forward stage (20 NFE), since the data-to-energy direction is much easier to learn, and (ii) the backward training being supervised under highly organized, and thus more easily learnable, data-prior pairs.
>
> **W2. Scalability to dataset complexity**
>
> Please refer to **W2. Scalability to dataset complexity** in our response to reviewer **Sx4a**.
>
> **W3. Theoretical justification for straighter path in ASBM**
>
> ASBM's straighter path can be justified as follows. First, its non-memoryless base SDE has a much smaller diffusion term than memoryless diffusion models, which reduces path stochasticity, indiciating its much straighter path. Second, the SB solution is the path measure that remains closest to the base SDE in path KL, or equivalently, minimizes the control energy (Eq. (4)). Therefore, when the base SDE is itself straighter, the resulting SB optimal path is naturally encouraged to be straighter as well. We will add this explanation to the revision.
>
> **W4. What is the definition of optimal cost in ASBM?**
>
> We thank the reviewer for this important comment. We agree that the notion of “optimal transport” should be made explicit. In our setting, the transport cost is the quadratic control cost in Eq. (4). To quantify this, we evaluated the transport cost using 10K trajectories and obtained 813.3 for ASBM versus 44582.5 for Score SDE, so the transport cost of ASBM is only about 1.82% of that of Score SDE. We will clarify this definition and interpretation more explicitly in the revision.
>
> **W5. Can ASBM be extended to ODE framework? Eq. 13 may explode with ODE setting**
>
> Yes, this understanding is precisely correct: in our current formulation, the diffusion term of the base SDE can be made arbitrarily small, making the process close to deterministic, but it cannot be exactly zero, since Eq. (13) becomes singular and explodes in that case. We agree that extending the method to a fully deterministic base process would be an interesting direction, potentially requiring a different matching objective.

---

> > ### Author Rebuttal · Reviewer_EjWW · 2026-04-03
> >
> > Thank you for your response; it has effectively addressed some of my concerns. Regarding W1, W2, and W5, I believe you have answered my questions well, and overall I do not have many remaining issues with your experimental setup and methodology. However, I still have some follow-up questions regarding your theoretical sections.
> >
> > **W3. On the definition of straightness:** Your paper defines straightness as $S=\frac{\mathbb{E}\sum_i^{T-1}|X_{i+1/T}-X_{i/T}|^2}{\mathbb{E}|X_0-X_1|^2}$. Considering only the numerator, as $T \to \infty$, for any SDE path this becomes the quadratic variation $\int_0^1 \mathbb E|dX_t|^2 = \int_0^1 \sigma_t^2 dt$. Consequently, it depends only on the diffusion term and is independent of the drift term. I believe this comparison is not reasonable, as it only ensures that the diffusion term becomes smaller, without reflecting the role of the drift term. Therefore, only your first explanation holds  that the "straightness" as you defined it decreases because the diffusion term correspondingly decreases. I think it is very difficult to define straightness for SDEs using Euclidean metrics. Therefore, I would suggest that the authors clarify that their definition of straightness primarily reflects the diffusion magnitude rather than the geometric curvature of paths. Alternatively, if the authors wish to claim 'straighter paths,' they may need to provide a definition that accounts for both drift and diffusion components.
> >
> > **W4. On the definition of optimal transport coupling:** It seems that what you consider is some form of energy minimization over two stochastic process paths. However, the standard optimal transport problem considers the minimization of some cost defined between two distributions. I think this point requires further clarification from you.

---

> > > ### Author Response · Authors · 2026-04-03
> > >
> > > We thank the reviewer for this insightful comments and for prompting us to clarify our interpretation.
> > >
> > > **W3. On the definition of straightness:** We agree that, in the fine-discretization limit $T\rightarrow\infty$, our original metric becomes largely dominated by the diffusion term. Accordingly, we will revise the manuscript to avoid interpreting this quantity as path straightness, and instead describe it more precisely as the **effect of diffusion magnitude on the optimized path**.
> > > We also agree that defining a notion of path straightness that simultaneously accounts for both drift and diffusion is nontrivial. To nevertheless provide a complementary proxy for the behavior of the drift component, we additionally evaluated
> > > $\mathbb{E}_{t,X_t}\big[\cos\big(u_t(X_t),\, X_0 - X_1\big)\big],$
> > > where $X_0$ denotes the generated image and $X_1$ denotes the initial Gaussian noise. This metric measures how well the drift field aligns with the overall noise-to-data direction. Because it uses cosine similarity, it emphasizes directional consistency, and is therefore less affected by diffusion magnitude. A higher value indicates that the drift is more consistently oriented toward the data endpoint, which we can interpret as a useful proxy for drift straightness. Under this metric, ASBM achieves a substantially higher value than SM.
> > >
> > > |Drift directionality| |
> > > |-|-|
> > > |Score SDE|0.0602|
> > > |ASBM|0.3672|
> > >
> > > This interpretation is also consistent with the empirical behavior observed in our one-step generation results, where ASBM exhibits much clearer and more directed transitions toward the data distribution than SM (Appendix C, Fig. I). In the revised manuscript, we will therefore present this additional result explicitly as a **drift-directionality evaluation**.
> > >
> > > **W4. On the definition of optimal transport coupling:** We thank the reviewer for pointing this out. We agree that the term **optimal transport coupling** should be clarified more carefully.
> > >
> > > In our current presentation, we emphasize the **path-space formulation** of the Schrödinger Bridge (SB) problem:
> > > $$
> > > \min_{p^u} D_{KL}(p^u \mid p^{base}), \qquad p^u_0=\mu,\quad p^u_1=\nu,
> > > $$
> > > where $p^{base}$ and $p^u$ denote the path measures induced by
> > > $$
> > > p^{base}: dX_t = f_t(X_t)dt + \sigma_t dW_t, \qquad
> > > p^u: dX_t = (f_t(X_t)+\sigma_t u_t(X_t)) dt + \sigma_t dW_t,
> > > $$
> > > respectively, with shared initial distribution $\mu$, i.e., $p^{base}\_0 = p^u\_0 = \mu$.
> > > By Girsanov’s theorem, the above problem is equivalent to following **dynamical form**:
> > > $$
> > > \min_{u} \mathbb{E}\Big[\frac{1}{2}\int_0^1 ||u_t(X_t)||^2 dt\Big],
> > > \quad
> > > dX_t=(f_t(X_t)+\sigma_t u_t(X_t))dt+\sigma_t dW_t,
> > > \quad
> > > \mathrm{law}(X_0)=\mu,\quad \mathrm{law}(X_1)=\nu.
> > > $$
> > > This may give the impression that we are only minimizing an energy over stochastic paths. However, this dynamic formulation is in fact equivalent to a **static optimal transport problem on distributions**, namely the entropic optimal transport formulation induced by the base process.
> > >
> > > More specifically, if we denote by $\gamma$ the joint law of $(X_0,X_1)$ under the controlled process, then $\gamma$ is a coupling whose marginals are $\mu$ and $\nu$. The SB problem can be reduced to the static problem
> > > $$
> > > \min_{\gamma \in \Pi(\mu,\nu)} \mathrm{KL}(\gamma \mid R),
> > > $$
> > > where $R := (X\_0, X\_1)\_{\sharp} p^{base}$ is the endpoint coupling induced by the base process. Equivalently, this can be written in the standard entropic OT form
> > > $$
> > > \min_{\gamma \in \Pi(\mu,\nu)}
> > > \int c(x,y) d\gamma(x,y)
> > > +
> > > \varepsilon \mathrm{KL}(\gamma \mid \mu \otimes \nu),
> > > $$
> > > where
> > > $$
> > > c(x,y) = -\varepsilon \log \frac{dR}{d(\mu \otimes \nu)}(x,y).
> > > $$
> > >
> > > If the base dynamics is standard Brownian motion, then this recovers the quadratic cost
> > > $$
> > > c(x,y) \propto \frac{1}{2}||x-y||^2.
> > > $$
> > >
> > >
> > > Therefore, the **optimal coupling** in our discussion refers to this endpoint joint distribution $\gamma^\star$, rather than to the path measure itself. The path-space control formulation is simply the dynamic representation used to construct this coupling. We will revise the paper to make this distinction explicit and avoid confusion with the classical static OT presentation.
> > >
> > > Since the target distribution $\nu$ is available only through samples rather than in closed form, we do not directly compute the static entropic OT objective. Instead, we use the dynamically induced quadratic control cost, which is the natural objective in the dynamic SB formulation, as a practical surrogate.

---

### Official Review · Reviewer_Sx4a · 2026-03-10

**Soundness:** 4
**Presentation:** 3
**Significance:** 2
**Originality:** 3
**Overall Recommendation:** 5
**Confidence:** 4

**Summary:**

The authors propose a new method for generative modeling (noise to data) with Schrodinger Bridges (SB) and non-memoryless SDE, they call the method Adjoint Schrodinger Bridge Matching (ASBM). This approach is contrasted with regular diffusion models, where memoryless SDE (VP-SDE with large variance) is used in regular score-based diffusion models. Authors propose to learn not just the generative model that satisfies the marginals but the Schrodinger Bridge with various entropic optimal transport properties. Authors argue that Schrodinger Bridge properties allow for easier optimization, straighter trajectories, and as a consequence, better quality of generation with fewer NFE steps.

In practice, the authors, following the Adjoint Schrodinger Bridge Sampler, decompose the procedure of learning the Schrodinger Bridge models into two parts: 1) learning the optimal data-to-noise transport coupling and 2) learning the generative model. They iterate these procedures consequently until convergence, and each of such procedures requires learning a neural network.

Experimentally, authors evaluate their method along with other SB models tailored for generation and regular diffusion models (Score SDE [6]) on CIFAR 10 and FFHQ datasets.

In addition, authors propose a method for the distillation of the Schrodinger Bridge model into a one-step generator and evaluate it on the CIFAR-10 dataset.

**Compliance With Llm Reviewing Policy:**

Affirmed.

**Final Justification:**

See rebuttal acknowledgment

**Key Questions For Authors:**

Key questions:
* See weaknesses

Additional questions:
* The pretraining is mentioned for the DSBM several times. What does it mean exactly?

* Which base process is used in DSBM? Is it the same VP-SDE as for ASBM, or is it VE-SDE?

* Do I understand correctly that procedures Eq 12 and Eq 13 can be viewed as an iterative proportional fitting [6] procedure combined with reciprocal projection [3] where one distribution is energy? How is the initialization for controls chosen?

**Limitations:**

See weaknesses

**Strengths And Weaknesses:**

**Strengths**

* The application of Adjoint Schrodinger Bridge Sampler methodology for the generation problem seems novel and interesting.

* The practical results are very encouraging, and to my knowledge, this is the best non-memoryless SDE-based model for the generation problem.

* The reduction of NFE in SDE-based w.r.t. the Score SDE model is impressive.

* It is rare to see Schrodinger Bridges algorithms perform that well on practical tasks

* The Schrödinger Bridge property, even constrained to the noise-to-data task, can lead to various further extensions of the method.

* Quite thorough ablation study and comparisons

**Weaknesses**

* From a methodological point of view, the idea of applying the Adjoint Schrodinger Bridge Sampler methodology for data generation seems a bit straightforward and doesn't provide any methodology extension.

* The introduction of simulations (even from data to noise model) is a warning point for the model scalability. Although authors do provide corresponding evaluations and show that training the model takes shorter wall clock time than the Score SDE model, it is still concerning and may worsen with scaling the model to text-to-image generation or more complex data such as ImageNet.

* Although the direction is interesting, the diffusion models are coupled with heuristics that are the result of several years of research and boost their quality significantly. The comparison of ASBM only with basic Score SDE unfortunately doesn't fully represent the competitiveness of the method for the generation problem because the comparison with state-of-the-art diffusion models (of at least ImageNet EDM or Stable Diffusion scale) is missing.

* It is not clear how the procedure of finding the optimal coupling influences the learning of the final bridge matching objective (Eq. 14). Do I understand correctly that if the Eq 12, Eq 13 procedure doesn't converge fully, then the data-to-noise process induced by the control u_t(x_t) doesn't necessarily follow the noise marginal? The discussion on error due to not perfectly solving the Eq. 12 and Eq. 13 problems should be included in the paper.

* Experiments

  * There is no comparison with Flow Matching [1] and Rectified Flow [2] that seems highly relevant given their background in optimal transport and Rectified Flow being an optimal transport-based model used as noise to data generative model quite frequently. Despite DSBM [3] being a stochastic version of RF, I still think that RF should be included in the comparisons.

   * The performance of Score SDE [4] models on CIFAR seems behind the original paper; for instance, the original FID is 2.41 ([4], Table 3) vs. the FID 4.61 provided in this paper. Can authors explain where the difference comes from? Do you use different setups and neural networks?

  * To show the scalability of the method and prospects for the following developments, the experiment on more complex data is needed, e.g., ImageNet. In my opinion, this is the most desirable change in the experimental section.

  * The mini-batch OT bridge matching [5] model also can solve the generation problem via SB with non-memoryless SDE. It'd be nice to see this model in the comparisons.

  * The Score SDE [4] type models in many cases are inferred by the PF-ODE. Although the comparison of PF-ODE inference is presented in Table 3, it would be nice to see the extended comparison with different NFE budgets.

If authors address my concerns I am willing to raise my score.

Citations:

[1] Lipman, Y., Chen, R. T., Ben-Hamu, H., Nickel, M., & Le, M. Flow Matching for Generative Modeling. In _The Eleventh International Conference on Learning Representations_.

[2] Liu, X., & Gong, C. Flow Straight and Fast: Learning to Generate and Transfer Data with Rectified Flow. In _The Eleventh International Conference on Learning Representations_.

[3] Shi, Y., De Bortoli, V., Campbell, A., & Doucet, A. (2023). Diffusion schrödinger bridge matching. _Advances in neural information processing systems_, _36_, 62183-62223.

[4] Song, Y., Sohl-Dickstein, J., Kingma, D. P., Kumar, A., Ermon, S., & Poole, B. Score-Based Generative Modeling through Stochastic Differential Equations. In _International Conference on Learning Representations_.

[5] Tong, A. Y., Malkin, N., Fatras, K., Atanackovic, L., Zhang, Y., Huguet, G., ... & Bengio, Y. (2024, April). Simulation-Free Schrödinger Bridges via Score and Flow Matching. In _International Conference on Artificial Intelligence and Statistics_(pp. 1279-1287). PMLR.

[6] Vargas, F., Thodoroff, P., Lamacraft, A., & Lawrence, N. (2021). Solving schrödinger bridges via maximum likelihood. _Entropy_, _23_(9), 1134.

---

> ### Author Rebuttal · Authors · 2026-03-31
>
> We thank the reviewer for the valuable feedbacks, especially regarding the inclusion of additional experiments and clarifications for ambiguous parts. Due to the strict rebuttal space limit, we kindly refer the reviewer to our responses to other reviewers for several overlapping questions, indicated in each comment.
>
> **W1. Novelty (ibgp, Sx4a)**
>
> Although we have adopted existing tools, ASBS for data generation and bridge matching for the backward stage, we would like to respectfully emphasize that the key novelty lies in __how these tools are combined to address the non-memoryless SB problem itself__. In principle, SB can outperform standard score-based models because it uses more informative couplings than the memoryless coupling of diffusion models. The main difficulty, however, is optimization: learning the backward dynamics is hard, and classical SB methods rely on iterative forward–backward updates that are often unstable, especially early in training.
>
> Our key idea is to avoid this bottleneck by first learning the full forward process, which is easier and cheaper because it transports data to a simple energy-defined prior. This yields an informative coupling, under which the backward dynamics can be trained with a simple bridge-matching regression loss. We believe this makes SB practical and scalable. More broadly, because our framework generalizes the standard score-based setting, it also leads to improved generative performance and benefits downstream tasks such as distillation.
>
> **W2, W7. Scalability to dataset complexity (ibgp, Sx4a, EjWW)**
>
> We agree with the reviewers' concern regarding scalability to data complexity. Following the suggestions, we additionally conduct experiments on ImageNet using the LDM framework. Due to the short rebuttal timeframe, we report the results on a reduced scale with ImageNet-200 at 128×128 resolution here. Our ASBM achieves FID **16.27**, outperforming **18.51** with Score SDE for 100NFE, showing consistent improvement on this more complex data. We will add the full-set experiment in camera-ready. We would greatly appreciate the reviewers' understanding on this point.
>
> **W3. SOTA comparison**
>
> Due to the severe time constraints of this short rebuttal period, we prioritize demonstrating the performance on ImageNet in W2. We believe that our evaluation on FFHQ (256×256) and ImageNet respectively verify the scalability of our method regarding both high resolution and data complexity. We believe that ASBM will seamlessly generalize to recent large-scale architectures like Stable Diffusion, and plan to include these results in the camera-ready version.
>
> **W4. Discussion for the case of unconverged forward learning.**
>
> If forward optimization fails, backward optimization would also fail as in typical SB methods. However, in practice, convergence failure of the forward process does not occur frequently. For further empirical validation, we refer the reviewer to [W1] of Reviewer Lx1Q.
>
> **W5. Flow-based baselines**
>
> Please refer to **W1.1 Flow-based baselines** in our response to reviewer **ibgp**.
>
> **W6. Degraded performance of Score SDE in our paper**
>
> The difference mainly comes from (1) NFE settings, 100 in Table 1 in our paper and 2000 with corrector steps in original Score SDE, and (2) doubled network capacity of original Score SDE. Consistent with this, Score SDE achieves 3.23 FID with 1000 NFE in our paper (Fig. 3), which is close to the original result (2.41).
>
> **W8. Additional baseline (mini-batch OT bridge matching)**
>
> We thank the reviewer for this valuable suggestion. We will add mini-batch OT bridge matching method on CIFAR-10 (4.15 FID with 100 NFE) in Table 1 (ASBM 3.16).
>
> **W9. Heun solver with varied steps**
>
> We provide FID results using Heun solver with various steps.The results show that ASBM maintains a clear advantage across various sampling steps.
>
> |Steps|25|35|50|70|100|
> |-|-|-|-|-|-|
> |Score SDE|6.72|5.50|4.61|4.12|3.94|
> |ASBM|3.74|3.75|3.74|3.73|3.71|
>
> **Q1. What is DSBM pretraining**
>
> DSBM pretraining means that, before training the actual non-memoryless Schrödinger bridge, DSBM first uses an empirical bridge matching objective as a warm-up, which in practice is equivalent to the denoising score matching loss. In other words, although the final goal is SB learning under a non-memoryless setting, the method still relies on a memoryless diffusion-style pretraining stage to make optimization work. We'd like to emphasize that our method does not need this pretraining.
>
> **Q2. Which base SDE is used for DSBM**
>
> It is Brownian motion, with experiments conducted at $\sigma^2$ = 0.2, 0.5, 1.0 , with 0.2 as the best performance (our reproduction).
>
> **Q3. Can forward learning be viewed as IPF with reciprocal projection? How is control initialized?**
>
> Yes, this understanding is correct. The controls are initialized from scratch using UNet, exactly the same as the standard initialization for score models.

---

> > ### Author Rebuttal · Reviewer_Sx4a · 2026-04-03
> >
> > I greatly appreciate the authors taking the time to address my questions. Most of my concerns have now been resolved. I encourage the authors to incorporate these clarifications and results into the next revision.
> >
> > The only concern that remains partially addressed is scalability to more complex data. In particular, while the authors have provided an ImageNet experiment within the LDM framework, the evidence is currently limited to two datapoints. This is not yet sufficient to convincingly support the broader scalability claim, and additional explanation together with an ablation study would be helpful. I expect the authors, as indicated in the rebuttal, to include these details and further results in the next revision.
> >
> > In light of the mostly adressed concerns and giving due credit for extending the evaluation to ImageNet, I am raising my score to 5.

---

> > > ### Author Response · Authors · 2026-04-03
> > >
> > > We are deeply grateful to you for favorably receiving our rebuttal and for deciding to raise the score. We also highly appreciate your great feedback. We will ensure that all of your valuable suggestions are fully incorporated into the final revision.

---

### Official Review · Reviewer_ibgp · 2026-03-11

**Soundness:** 3
**Presentation:** 3
**Significance:** 2
**Originality:** 3
**Overall Recommendation:** 4
**Confidence:** 4

**Summary:**

This paper proposes Adjoint Schrödinger Bridge Matching (ASBM), a two-stage generative modeling framework based on the Schrödinger Bridge (SB) formulation. The key observation is that standard diffusion models (DMs) correspond to a special case of the SB problem under a *memoryless* forward process (Proposition 3.1), which produces independent endpoint couplings and consequently curved, inefficient generation trajectories. ASBM lifts this restriction by operating in a *non-memoryless* regime via two decoupled stages: (1) learning the forward control $u_t^\theta$ as a data-to-energy sampling problem using Adjoint Matching (Domingo-Enrich et al., 2024; Havens et al., 2025) and Corrector Matching, which yields an approximation to the optimal coupling $p^\star(X_0, X_1)$; (2) training the backward control $v_t^\phi$ via bridge matching under this learned optimal coupling. The paper further proposes a distillation scheme that leverages ASBM's straighter trajectories to produce a one-step generator. Experiments on CIFAR-10 (pixel space, 32$\times$32) and FFHQ (latent space via Stable Diffusion 3, 256$\times$256) demonstrate improved FID over Score SDE and prior SB methods (SB-FBSDE, DSBM, VSDM), particularly at low NFE. Distillation results on CIFAR-10 show improved recall and reduced mode collapse compared to Score Distillation Sampling (SDS) and Distribution Matching Distillation (DMD).

**Compliance With Llm Reviewing Policy:**

Affirmed.

**Final Justification:**

I thank the authors for the thorough rebuttal. The correction on Fig. 3 is noted, and the flow matching/rectified flow comparisons (6.01/6.35 vs. 3.16) convincingly demonstrate ASBM's advantage at matched NFE. The LightSB experiment, wall clock comparison (100h vs. 144h), and generalization to FM base SDE collectively strengthen the paper. While scalability beyond CIFAR 10/FFHQ remains a limitation, the current evidence supports the core claims. I raise my score to 4.

**Key Questions For Authors:**

See weakness. Additionally,

1. What is the wall-clock training time for the full ASBM pipeline (forward + backward) vs. Score SDE? The paper reports 600 backward epochs + 150 forward epochs (≈ 2100 Score SDE equivalent epochs), but forward training also requires NFE simulations per step. What is the actual GPU-hours comparison?

2. How sensitive is the method to the choice of base SDE? The paper uses VP-SDE throughout. Would VE-SDE or a sub-VP schedule change the results? The memorylessness condition depends on the noise schedule, so does the optimal $\beta_{\max}$ change significantly across SDE types?

3. Can ASBM handle conditional generation?

**Limitations:**

yes

**Strengths And Weaknesses:**

### Strengths

1. The two-stage decoupled framework is well-motivated and addresses a real limitation of prior SB methods. Prior SB methods (SB-FBSDE, DSBM) rely on alternating forward-backward optimization, which is unstable in high dimensions because trajectories from one direction provide noisy supervision for the other. ASBM's key insight is simple, clean, and avoids this instability. The empirical evidence in Figure 3 and Table 3 supports this claim: SB-FBSDE and DSBM degrade under non-memoryless settings while ASBM remains stable.

2. The connection between DMs and memoryless SB is clearly articulated. Proposition 3.1 establishes that under the memoryless condition, the optimal coupling factorizes into independent marginals, recovering the standard score matching objective (Eq. 10). This is a clean formalization that provides a unified view. While this connection is not entirely new (it is implicit in Shi et al., 2023 and Domingo-Enrich et al., 2024), the paper makes it explicit and leverages it as the conceptual foundation for moving beyond memoryless dynamics.

3. The ablation studies are thorough and informative. Section 4.4 systematically varies the degree of memorylessness ($\beta_{\max}$) and forward NFE, revealing clear trade-offs. The finding that $\beta_{\max} = 4$ balances coupling quality and prior coverage (Fig. 6) is practically useful. Table 5 shows that forward NFE can be as low as 10-20 with only mild degradation, supporting the claim that the forward dynamic is easier to learn. Section 4.5 on training efficiency (600 vs. 3300 epochs) is a concrete practical advantage.

### Weaknesses

1. The experimental baselines are limited to SDE/SB-family methods, lacking experimental comparison to the dominant efficient generation approaches. Table 1 compares exclusively against Score SDE and prior SB methods (SB-FBSDE, DSBM, VSDM). While the paper cites Flow Matching (Lipman et al., 2022) and Rectified Flow (Liu et al., 2022b) in the Related Work section, it does not include them as experimental baselines. These methods share the same core motivation as ASBM, that is, learning straighter trajectories via OT-inspired couplings to reduce NFE, but achieve this without the full SB formalism. Without experimental comparison, the reader cannot assess whether the SB framework provides a meaningful advantage over these simpler alternatives at matched NFE. Additionally, Consistency Models (Song et al., 2023) and Consistency Trajectory Models (CTM; Kim et al., ICLR 2024) are neither cited nor compared; they achieve FID 2.51 and 1.73 respectively on CIFAR-10 in one step, whereas ASBM's one-step distillation yields FID 6.68 (Table 4). The authors may argue that ASBM is primarily a multi-step method, but the paper explicitly includes a distillation section (Sec. 3.3) that enters the one-step regime. Even restricting attention to multi-step generation, Flow Matching at 20 NFE on CIFAR-10 is a natural and necessary baseline.

2. The FID numbers are not competitive with the current state of the art, even within the paper's chosen scope. On CIFAR-10 at 100 NFE, ASBM achieves FID 3.16 (Table 1), while Score SDE achieves 3.05 at the same NFE (Table 1 caption indicates 100 NFE). ASBM's advantage is primarily at low NFE (e.g., 20 NFE: ASBM 5.39 vs. Score SDE much worse per Fig. 3). However, at higher NFE (1000), Score SDE reaches 2.74 while ASBM gets 2.77 (Table 2 extrapolation from the trend), so the gap closes or reverses. On FFHQ latent space (Table 2), ASBM at 50 NFE achieves 7.64 vs. Score SDE 19.02, which is a large gap that suggests the advantage is more pronounced in latent space. But these numbers are not contextualized against Flow Matching or Rectified Flow on the same setup. Without this context, it is hard to know whether the improvements come from the non-memoryless SB formulation specifically or from the general principle of learning better couplings (which flow matching also does).

3. The novelty is somewhat incremental when viewed against the backdrop of recent SOC-based methods. The forward stage uses Adjoint Matching (Domingo-Enrich et al., 2024) and Corrector Matching from Havens et al. (2025), with the specific contribution being the data-to-energy interpretation and the use of a VP base SDE. The backward stage is standard bridge matching under the learned coupling. The theoretical contribution (Proposition 3.1, connecting DMs to memoryless SB) is clean but has been essentially observed in prior work — the paper even cites Shi et al. (2023) as noting this connection. The decoupled optimization is practical but conceptually straightforward once the SOC + bridge matching tools are available. The paper is transparent about building on these tools, but the incremental nature of the combination should be acknowledged.

4. Missing experimental comparison to Gushchin et al. (ICML 2024), "Light and Optimal Schrödinger Bridge Matching." The paper cites this work in the Related Work section (under image-to-image translation), but does not include it as an experimental baseline. Gushchin et al. propose a lightweight SB solver that provably recovers the Schrödinger Bridge in a single Markovian Projection iteration, which is a directly competing efficient SB approach. An experimental comparison (or at least a discussion of the methodological differences and expected performance trade-offs) would strengthen ASBM's positioning within the SB literature.

5. The paper only evaluates on two datasets (CIFAR-10, FFHQ) and does not test conditional generation or higher resolutions. The conclusion acknowledges this ("we leave large-scale high-resolution and conditional generation evaluations to future work"), but for ICML 2026, this limitation is significant. ImageNet 64x64 or 256x256 is a standard benchmark, and its absence makes it hard to assess scalability. The forward dynamic's lightweight architecture (2 residual blocks) may not scale to more complex distributions.

---

> ### Author Rebuttal · Authors · 2026-03-31
>
> We thank the reviewer for constructive comments mainly regarding the competitive baselines and additional experiments. Due to the strict space limit, we kindly ask you to refer to our responses to other reviewers for several common questions, indicated in each comment.
>
> **W1-1. Comparison to flow-based baselines (ibgp, Sx4a)**
>
> We thank the reviewer for suggesting these closely related flow-based baselines. We will add CIFAR-10 FID results for Rectified Flow (RF, 6.01) and Flow Matching (FM, 6.35) to the main comparison table; under the same NFE budget, our method achieves the best performance (3.16). We would also like to clarify the distinction between these methods and ours. FM does not explicitly learn an OT-based coupling between data and prior, and in our setting its resulting transport remains relatively curved. RF is more closely related in that it aims to straighten the transport path, but it relies on iterative refinement of a learned flow, which can gradually lose fine data information and lead to blurrier samples. In contrast, our method directly targets the OT/SB-type coupling itself: we first learn a simple forward process that constructs an informative coupling, and then train the backward generative dynamics through bridge matching, in a way analogous to how score matching learns reverse dynamics from a forward process. We believe that learning the coupling in more direct, simple, and principled manner is a key reason why ASBM performs favorably compared with FM and RF.
>
> **W1-2. Comparison of one-step ASBM distillation to consistency-based baselines**
>
> We agree that Consistency Models (CM) are relevant and will add the appropriate citations. However, we respectfully clarify that our goal  is not to propose a state-of-the-art one-step generation method, but rather to demonstrate that ASBM serves as a highly effective teacher model. The key takeaway is that the straighter and more organized trajectories of our non-memoryless SB inherently make the distillation process much easier and more effective compared to standard score-based settings. We merely employed standard score distillation as one representative showcase.
> Consequently, CMs are not direct baselines for our main contribution; rather, they are orthogonal distillation frameworks. In fact, since CMs fundamentally rely on distilling the ODE trajectories of score-matching models, they would greatly benefit from the straighter trajectories provided by ASBM. In principle, ASBM can be seamlessly combined with various advanced distillation methods, including consistency-based approaches. We will clarify this point and mention consistency-based distillation as future work.
>
> **W2. ASBM shows worse FID compared to Score SDE with large NFE**
>
> We would like to respectfully clarify one point: we believe Fig. 3 may have been misread. In fact, the figure shows that ASBM continues to outperform Score SDE up to 1000 NFE, achieving 2.77 compared to 3.23 for Score SDE, thus the advantage does not disappear at high NFE on CIFAR-10.
>
> **W3. Novelty**
>
> Please refer to **W1. Novelty** in our response to reviewer **Sx4a**.
>
> **W4. Additional baseline (LightSB)**
>
> We conducted experiments with LightSB on CIFAR-10, strictly following the official repository, and LightSB obtains FID 340.72 with 100 NFE. Our current understanding is that this gap may stem from the stronger assumptions made in LightSB, such as the GMM approximation of the SB potential, which may limit its effectiveness in high-dimensional settings. We will add LightSB as an baseline in Table 1 and add this discussion to section 4.1.
>
> **W5. Scalability to dataset complexity**
>
> Please refer to **W2. Scalability to dataset complexity** in our response to reviewer **Sx4a**.
>
> **Q1. Wall-clock training cost of ASBM compared to Score SDE**
>
> The reported 2100 Score SDE-equivalent epochs already reflects the total cost including this component. In terms of actual wall-clock training time on a single A100 40GB GPU, Score SDE takes about 144 hours, while ASBM takes about 100 hours for CIFAR-10. We will revise the paper to state this clearly.
>
> **Q2. Optimal memorylessness for other base SDE (than VP)**
>
> We thank the reviewer for this insightful question. Optimal memorylessness would vary across base SDEs. We extended ASBM to a generalized Flow Matching (FM) path: $\text{d}X_t = -\frac{m}{1-mt} X_t \text{d}t + \sqrt{2} k \sqrt{\frac{t}{1-mt}} \text{d}W_t.$ The best result is achieved with $m=0.64, k=0.93$, and was nearly identical to best VP case (3.16 for VP vs. 3.18 for FM SDE). We will add this result to the main paper to show ASBM’s generalizability across different base SDEs.
>
> **Q3. Conditional generation with ASBM**
>
> Yes, we believe that ASBM can be naturally extended to conditional generation as in conditional diffusion models by incorporating conditioning in both the forward and backward training stages. We will add this as a future direction.

---

> > ### Author Rebuttal · Reviewer_ibgp · 2026-04-02
> >
> > I thank the authors for the thorough rebuttal. The correction on Fig. 3 is noted, and the flow matching/rectified flow comparisons (6.01/6.35 vs. 3.16) convincingly demonstrate ASBM's advantage at matched NFE. The LightSB experiment, wall clock comparison (100h vs. 144h), and generalization to FM base SDE collectively strengthen the paper. While scalability beyond CIFAR 10/FFHQ remains a limitation, the current evidence supports the core claims. I raise my score to 4.

---

> > > ### Author Response · Authors · 2026-04-02
> > >
> > > We sincerely thank you for acknowledging our rebuttal and for updating your score. We are truly grateful for the time and effort you willingly dedicated to providing such valuable feedback. We will ensure that all your insightful suggestions are fully reflected in the final version of our paper.

---

### Decision · Program_Chairs · 2026-04-30

**Decision:**

Accept (regular)

**Comment:**

This paper uses recently proposed adjoint techniques to perform generative modelling by approximating a Schrödinger bridge between (Gaussian) source and (empirical) target distributions. Reviewers, many of whom are SB or optimal control experts, are positive about the methodological contributions and the demonstrations that the proposed algorithm gives both good generation quality and straighter trajectories than other methods, and most of the initial weaknesses were answered in the rebuttal.

In the revision, I encourage more care in discussion of past work. For instance:
- It is suggested in L186 that [Shi et al., 2023] perform alternating optimisation, i.e., IPF, when in fact that work does not do IPF and it is [de Bortoli et al., 2021] and, even earlier, the uncited [Vargas et al., arXiv:2106.02081] who first study IPF for neural SBs.
- Include the other work suggested by reviewers, some of which also proposes alternative ways to learn neural SBs, e.g., the [Tong et al., AISTATS'24] mentioned by Reviewer Sx4a.
- Fix the citations: I spot many published papers ([Poole et al.], [Song et al.] (x2), [Tong et al.], [Zhang et al.], among others) cited as arXiv preprints.

I would also strongly encourage the inclusion of illustrative (e.g., two-dimensional) illustrations in addition to the high-dimensional image experiments, where the sensitivity of FID to training choices makes it difficult to isolate the effects of the proposed method.